# Cocaine chemogenetics blunts drug-seeking by synthetic physiology

Juan L. Gomez[1,6], Christopher J. Magnus[2,3,4,6], Jordi Bonaventura[1,5], Oscar Solis[1], Fallon P. Curry[1], Marjorie R. Levinstein[1], Reece C. Budinich[1], Meghan L. Carlton[1], Emilya N. Ventriglia[1], Sherry Lam[1], Le Wang[3], Ingrid Schoenborn[1], William Dunne[1], Michael Michaelides[1✉] & Scott M. Sternson[2,3,4✉]

Chemical feedback is ubiquitous in physiology but is challenging to study without perturbing basal functions. One example is addictive drugs, which elicit a positive-feedback cycle of drug-seeking and ingestion by acting on the brain to increase dopamine signalling[1–3]. However, interfering with this process by altering basal dopamine also adversely affects learning, movement, attention and wakefulness[4]. Here, inspired by physiological control systems, we developed a highly selective synthetic physiology approach to interfere with the positive-feedback cycle of addiction by installing a cocaine-dependent opposing signalling process into this body–brain signalling loop. We used protein engineering to create cocaine-gated ion channels that are selective for cocaine over other drugs and endogenous molecules. Expression of an excitatory cocaine-gated channel in the rat lateral habenula, a brain region that is normally inhibited by cocaine, suppressed cocaine self-administration without affecting food motivation. This artificial cocaine-activated chemogenetic process reduced the cocaine-induced extracellular dopamine rise in the nucleus accumbens. Our results show that cocaine chemogenetics is a selective approach for countering drug reinforcement by clamping dopamine release in the presence of cocaine. In the future, chemogenetic receptors could be developed for additional addictive drugs or hormones and metabolites, which would facilitate efforts to probe their neural circuit mechanisms using a synthetic physiology approach. As these chemogenetic ion channels are specific for cocaine over natural rewards, they may also offer a route towards gene therapies for cocaine addiction.

The addictive properties of drugs are dependent on the temporal dynamics of their brain exposure, which is strongly affected by dose, route and reinforcement schedule[5–7]. Traditional approaches for addiction research and therapy use open-loop neural perturbations, such as pharmacology and chemogenetics[8], that are not directly coupled to the addictive drug brain concentrations, and even fast-timescale optogenetic or electrical stimulation are not yoked to changes in addictive drug exposure levels[9,10]. As the neural signalling pathways from addictive drugs are sensitive to the drug-exposure time course, it is desirable to tailor neural perturbations to the temporal dynamics of addictive drug exposure. To address this limitation of open-loop techniques, we set out to develop specialized closed-loop chemogenetic receptors gated by specific addictive drugs. This is an approach to alter neuron activity in a manner that precisely mirrors the variable dynamics of addictive drug ingestion and brain pharmacokinetics. Inspired by physiological systems, we envisioned that engineered receptors could be used to create artificial opposing signalling processes between drug ingestion and drug reinforcement to selectively

suppress drug-seeking (Fig. 1a) without affecting responses for natural rewards.

## Cocaine-activated ion channels

We initially focused on chemogenetic receptors for cocaine to add to the tools available for investigating psychostimulant mechanisms, which may also have implications for treating cocaine addiction. The reinforcing effects of cocaine are mediated by blockade of the dopamine reuptake transporter (DAT, encoded by *Slc6a3*), leading to elevated extracellular dopamine levels[2]. Cocaine-activated chemogenetic receptors would allow individual neural populations to be systematically examined for their role in sustaining cocaine-seeking. Cocaine is an especially challenging molecule for developing chemogenetic receptors because DAT is a cocaine-antagonized co-transporter protein not readily adaptable to creating a cocaine-gated ion channel that can directly modulate neural activity. However, cocaine has an amine-containing pharmacophore that is chemically analogous to

[1]Biobehavioral Imaging and Molecular Neuropsychopharmacology Section, National Institute on Drug Abuse Intramural Research Program, Baltimore, MD, USA. [2]Howard Hughes Medical Institute; Janelia Research Campus, Ashburn, VA, USA. [3]Department of Neurosciences, University of California San Diego, La Jolla, CA, USA. [4]Howard Hughes Medical Institute, University of California San Diego, La Jolla, CA, USA. [5]Departament de Patologia i Terapèutica Experimental, Institut de Neurociències, Universitat de Barcelona, Barcelona, Spain. [6]These authors contributed equally: Juan L. Gomez, Christopher J. Magnus. ✉e-mail: mike.michaelides@nih.gov; ssternson@ucsd.edu

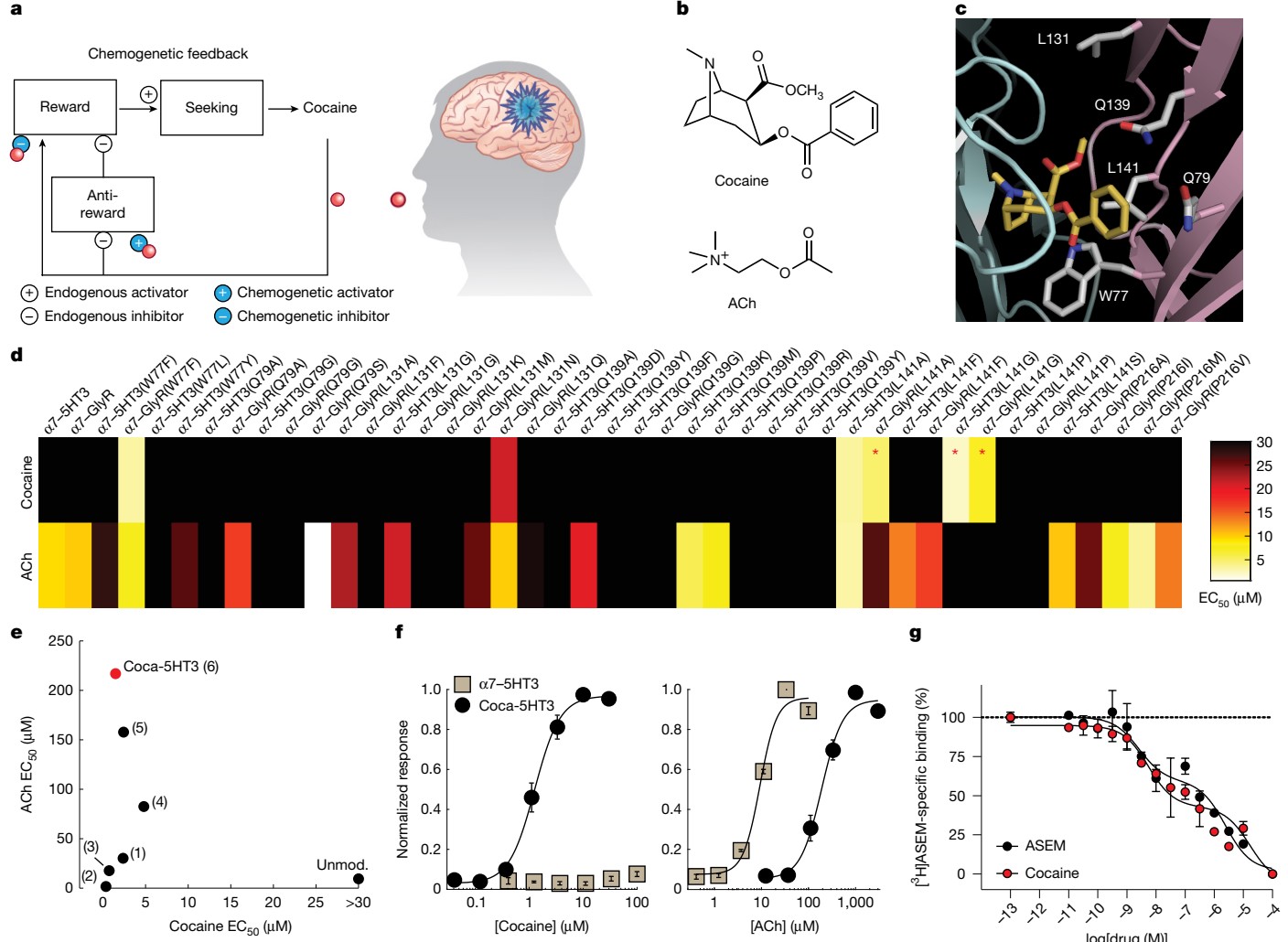

**Fig. 1 | Cocaine-gated ion channels. a**, Synthetic physiology schematic for using cocaine chemogenetics to reverse reinforcement after ingestion of an addictive drug that binds to a chemogenetic receptor targeted to a neural circuit. The diagram was adapted with permission from ref. 12, AAAS. **b**, The chemical structures of cocaine and ACh. **c**, Cocaine (yellow) binds to AChBP at the interface of two protomer subunits (cyan and pink). Homologous α7 nAChR pre-protein residues and numbering are shown (modified from Protein Data Bank (PDB): 2PGZ). **d**, The potency of cocaine and ACh agonism of chimeric channels comprising single-mutation α7 nAChR LBDs and IPDs from either 5HT3 or GlyR. The red asterisks highlight chimeric channels with cocaine agonism and reduced ACh potency. **e**, The relationship between ACh potency and cocaine potency at α7–5HT3 mutant chimeric channels: L141G (1), L141G G175K (2), L141G G175K Y217F (3), L141G G175K Y115F (4), L141G G175K Y210F (5), L141G G175K Y210F Y217F (6). Unmod., unmodified α7–5HT3. **f**, Coca-5HT3 potency for cocaine (left, *n* = 8 separate dose–responses) and ACh (right, *n* = 8 separate dose–responses) compared with unmodified α7–5HT3 (*n* = 3 separate dose–responses each). **g**, Displacement of [³H]ASEM at coca-5HT3 by cocaine or ASEM. *n* = 3 replicate curves per drug. Data are mean ± s.e.m.

molecules that have agonist activities at ion channels. Thus, we pursued a protein-engineering approach to develop cocaine-activated ion channels.

α7 nicotinic acetylcholine (ACh) receptors (nAChRs) are ligand-gated ion channels, and they have been engineered to possess new pharmacological properties by altering the ligand-binding domain (LBD)[11,12]. The LBD of α7 nAChR can be spliced to the ion pore domain (IPD) of the cation-selective serotonin receptor or the anion-selective glycine receptor, and the resultant α7–5HT3 or α7–GlyR chimeric channel subunits self-assemble into pentameric ion channels for neuron activation or inhibition, respectively[11–14] (Extended Data Fig. 1a). Importantly, agonists for nAChRs have pharmacophores with structural similarity to addictive drugs (Extended Data Fig. 1b). Cocaine (Fig. 1b) is a low-affinity antagonist of multiple nAChR subtypes[15–17]. A crystal structure of the homologous ACh-binding protein (AChBP)[18] in a complex with cocaine (Fig. 1c) shows a binding mode similar to canonical nAChR agonists, such as nicotine. Moreover, tropisetron, a molecule with even greater structural similarity to cocaine that shares the *N*-methyl tropane ring

structure (Extended Data Fig. 1b), is an agonist of α7 nAChR[19,20] as well as the analogous chimeric channels α7–5HT3 and α7–GlyR[12]. We therefore reasoned that the carboxymethyl ester of cocaine (Extended Data Fig. 1b) may be preventing cocaine agonist activity by colliding with bulky amino acids on the complementary face of the ligand-binding pocket that is formed at the protein–protein interface of two α7 nAChR LBD subunits. We investigated whether we could identify mutations that would both enhance cocaine affinity for the α7 nAChR LBD as well as confer novel cocaine agonist activity for the corresponding chimeric ion channels.

To identify mutated ion channels with cocaine agonist activity, we used a fluorescence membrane potential assay to screen the activity of chimeric α7–5HT3 and α7–GlyR ion channels with amino acid substitutions in the LBD that are in proximity to cocaine (Fig. 1d). Trp77Phe and Leu131Asn showed some cocaine agonist activity but also had undesired activity with the endogenous agonist ACh. Substitution of Leu141 with the smaller amino acid residues Gly or Ala produced a marked induction of cocaine agonism and reduced ACh potency. The bulky side chain

of Leu141 is proximal to the carboxymethyl ester of cocaine (Fig. 1c), indicating that reduction of steric clash in the ligand-binding pocket promotes cocaine binding and agonist activity.

To develop a cocaine-based chemogenetic platform for use in vivo, we performed additional optimization of the LBD to further improve cocaine potency and reduce ACh potency. ACh binding is facilitated by cation–pi interactions in the electron rich 'aromatic cage' from multiple Tyr residues[21–23]. A reduction in the electron density by mutation of Tyr210 and Tyr217 (Extended Data Fig. 1c) each to Phe right-shifted the ACh potency but also reduced the cocaine potency (Fig. 1e,f). The additional mutation of Gly175Lys (Extended Data Fig. 1c), which was previously reported to increase nicotine binding[24], improved the cocaine potency at α7(L141G,G175K,Y210F,Y217F)–5HT3, with the cocaine half-maximum effective concentration ($EC_{50}$) = 1.5 ± 0.3 μM (Fig. 1f). This $EC_{50}$ is below brain cocaine concentrations (~2.5–20 μM) estimated from self-administration[6,25]. The channel showed low potency activity of the endogenous nAChR agonists, ACh and choline (ACh $EC_{50}$ = 216 ± 35 μM, choline $EC_{50}$ = 1,212 ± 61 μM; Fig. 1e and Extended Data Fig. 2a), which was about 100-fold above the peak physiological levels (1–3 μM and 10–20 μM, respectively)[26,27]. Cocaine binding to α7(L141G,G175K,Y210F,Y217F)–5HT3 was also measured by displacement of the α7 nAChR antagonist [³H]ASEM[28] (2 nM). This showed a biphasic binding curve with 45 ± 10% of high-affinity binding sites (high-affinity inhibition constant ($K_{iH}$) = 1.6 ± 0.9 nM, low-affinity inhibition constant ($K_{iL}$) = 10 ± 7 μM) (Fig. 1g). The $K_{iH}$ of α7(L141G,G175K,Y210F,Y217F)–5HT3 for cocaine is greater than the range of cocaine affinities (0.23–2.0 μM) reported for its endogenous target, DAT[29–31]. We named this cocaine-gated cation channel coca-5HT3.

Importantly, coca-5HT3 was not activated by the main cocaine metabolites (ecgonine and benzoyl ecgonine; Extended Data Fig. 2b), probably due to unfavourable interactions with the free carboxylate group in these molecules. A panel of other addictive drugs that have an amine pharmacophore (amphetamine, methamphetamine, morphine, heroin and oxycodone) did not activate this receptor (Extended Data Fig. 2c). Even nicotine, which is an agonist of the unmutated α7–5HT3 chimeric channel, showed greater than 300-fold reduced potency for coca-5HT3 (Extended Data Fig. 2c,d). Finally, coca-5HT3 was not activated by a larger panel of endogenous signalling molecules at physiologically relevant concentrations (Extended Data Fig. 3a). Thus, coca-5HT3 exhibited highly selective pharmacology for cocaine, with no or low activity for other physiologically relevant amine-containing molecules that we tested. Some clinically used drugs have a tropane pharmacophore similar to cocaine, but most showed no or low potency activity at coca-5HT3 (Extended Data Fig. 3b).

Coca-5HT3 showed robust cocaine-activated currents in whole-cell voltage-clamp recordings in HEK293 cells in which the steady-state window currents associated with prolonged agonist applications corresponded to the $EC_{50}$ values from the membrane potential assay (Fig. 2a). The electrophysiological response to ACh (Fig. 2b) was shifted above physiological concentrations.

To examine neuromodulation by these cocaine-gated channels, we transfected cultured hippocampal neurons with vectors that co-expressed the fluorescent GFP reporter (pCAG::coca-5HT3-IRES-GFP). Current-clamp recordings of the coca-5HT3 cation channel in cultured hippocampal neurons depolarized neurons in a dose-dependent manner (Fig. 2c) but did not affect control neurons. Neuronal firing was increased at cocaine concentrations of 1–3 μM (Fig. 2d), bracketing the $EC_{50}$ of the mutated chimeric ion channel. Consistent with this, the amplitude of current required to elicit an action potential (rheobase) was reduced by 3 μM cocaine (Fig. 2e).

We obtained analogous results for a corresponding cocaine-gated chemogenetic chloride channel by swapping a similar α7 nAChR LBD (Leu141Gly, Gly175Lys and Tyr217Phe) onto the IPD of the glycine receptor (hereafter, coca-GlyR) (Fig. 2f–h). This channel showed a biphasic

cocaine competition binding curve with 37.9 ± 0.5% high-affinity binding sites ($K_{iH}$ = 0.014 ± 0.01 nM; $K_{iL}$ = 850 ± 500 nM) (Extended Data Fig. 4). Functional analysis showed large, sustained currents in whole-cell voltage-clamp recordings in transfected HEK293 cells (Fig. 2f,g). Coca-GlyR had a cocaine $EC_{50}$ of 1.2 ± 0.1 μM ($n$ = 7) and an ACh $EC_{50}$ of 63 ± 12 μM ($n$ = 8) (Fig. 2h). Cultured hippocampal neurons transduced with coca-GlyR fired normally in response to current injection, but firing was inhibited by cocaine (Fig. 2i), which was associated with a dose-dependent reduction of membrane input resistance (Fig. 2j). Correspondingly, the rheobase was increased (Fig. 2k). The passive membrane electrical properties of control neurons and neurons expressing coca-5HT3 or coca-GlyR did not show statistically significant differences (Fig. 2l).

## Cocaine chemogenetics blunts self-administration

We next tested the use of cocaine chemogenetics to counteract the reinforcement processes modulated by cocaine. The lateral habenula (LHb) has been reported to have an anti-reward function that blunts excessive reward-seeking[32] by activating rostromedial tegmental nucleus (RMTg) GABA neurons and ventral tegmental area (VTA) GABA neurons that both inhibit VTA dopamine neurons[33] (Fig. 3a). The LHb is normally inhibited by cocaine in vivo, in part through elevated dopamine acting on the dopamine type 2 (D2) receptor in LHb neurons[34], and LHb inhibition promotes cocaine self-administration[35]. Cocaine-mediated LHb inhibition is transient (0–15 min)[34], tracking the similarly short brain-exposure time course of cocaine[6], and is associated with reward in this timeframe[6,34,36]. We reasoned that an artificial opposing response produced by expressing coca-5HT3 excitatory channels in LHb neurons would reverse the sign of the cocaine-elicited inhibition response at this addiction node (Fig. 3a). This operation enabled us to install an artificial cocaine-dependent signalling node in the brain and to test the capacity of direct cocaine-mediated activation of LHb neurons to interfere with cocaine reinforcement solely during the dynamic drug-exposure time course after cocaine self-administration.

One of the most clinically predictive animal models for assessing the potential abuse liability of a drug is the intravenous (i.v.) drug self-administration (IVSA) procedure[37–39]. In the IVSA model of addiction, voluntary cocaine intake is dependent on the infusion dose, which approximates an inverted U-shaped curve. Interventions in animal models that shift down the dose–response curve are associated with clinical efficacy for addiction therapies, whereas treatments that right-shift the dose–response curve are more likely to result in compensatory increased drug administration[37–39] (Fig. 3b).

The LHb was targeted in rats by stereotaxic injection of the adeno-associated virus AAV5-Syn::coca-5HT3-IRES-mCherry (Fig. 3c,d). Control rats had sham surgery without AAV injection. LHb neurons expressing coca-5HT3 had similar basal electrophysiological properties to control neurons in the absence of cocaine (Extended Data Fig. 5). Rats were initially trained for food self-administration. All of the rats showed acquisition of food self-administration and rats injected with coca-5HT3 did not differ significantly from the controls (Fig. 3e) (two-way repeated-measures analysis of variance (RM-ANOVA), transgene main effect ($F_{1,24}$ = 2.21, $P$ = 0.15), transgene × session interaction ($F_{9,216}$ = 0.52, $P$ = 0.85)), showing that food-seeking was unaffected by coca-5HT3 expression in the LHb. Next, an i.v. catheter was implanted into the jugular vein of both experimental groups, and the rats were exposed to the cocaine IVSA procedure using a reinforcing dose of cocaine (0.5 mg per kg per infusion). We found that rats expressing coca-5HT3 and control rats acquired cocaine IVSA similarly (Fig. 3f) (two-way RM-ANOVA, transgene × session interaction, $F_{9,216}$ = 1.17, $P$ = 0.31). This shows that coca-5HT3 does not significantly interfere with cocaine self-administration using a high infusion dose that does not require high instrumental responding to maintain exposure to reinforcing levels of cocaine.

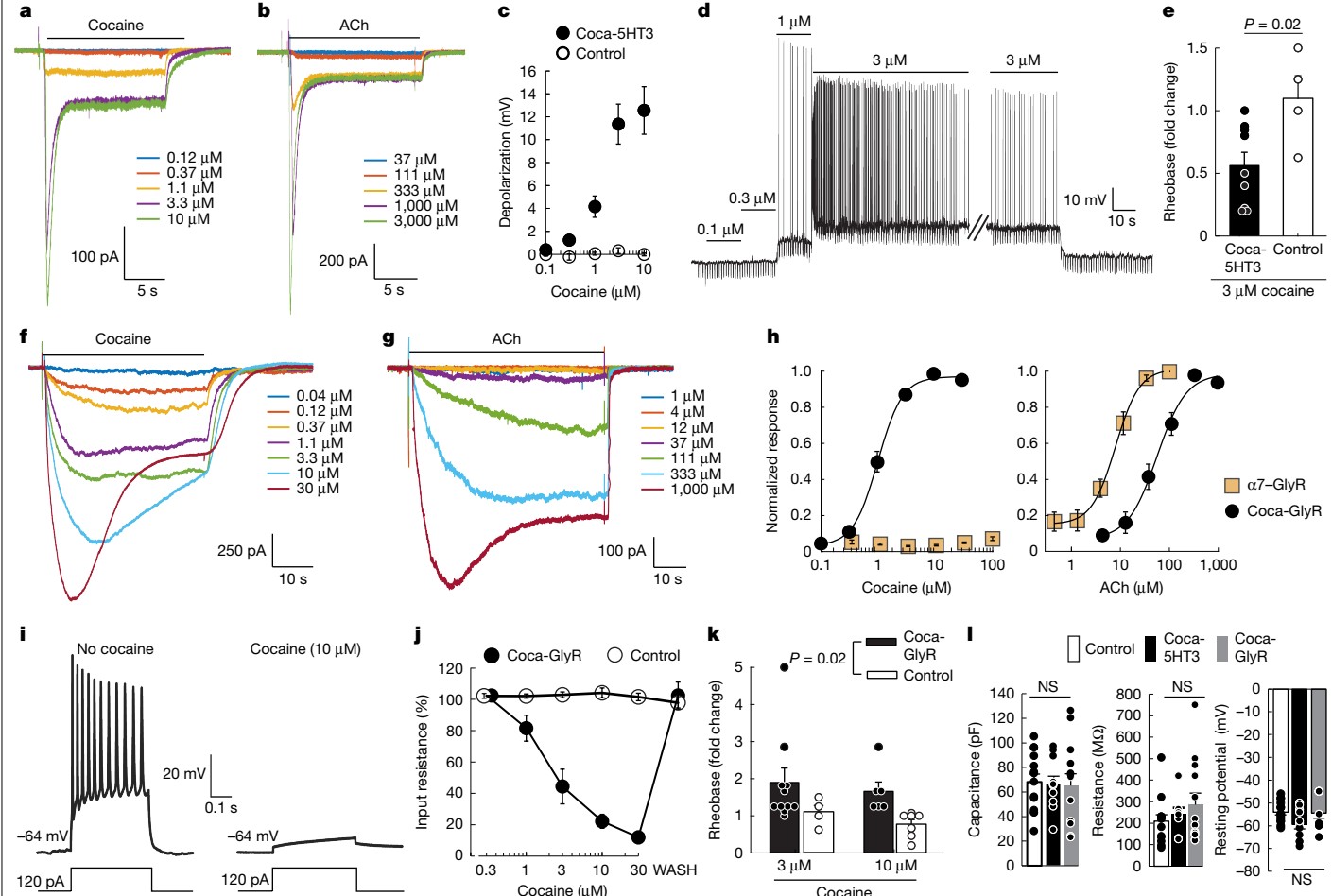

**Fig. 2 | Functional properties of excitatory and inhibitory cocaine-gated ion channels. a,b**, Cocaine-activated (**a**) and ACh-activated (**b**) currents from coca-5HT3. **c**, Depolarization in hippocampal neurons expressing coca-5HT3 ($n = 11$ cells) or GFP ($n = 9$ cells). **d**, Action-potential firing in a hippocampal neuron expressing coca-5HT3 in response to cocaine. Downward deflections indicate current injections to monitor membrane properties. **e**, The fold change in current necessary to elicit an action potential (rheobase) in neurons expressing coca-5HT3 ($n = 9$ cells) or GFP ($n = 4$ cells) in response to 3 μM cocaine. Statistical analysis was performed using a two-sided $t$-test; $P = 0.023$. **f,g**, Cocaine-activated (**f**) and ACh-activated (**g**) currents from coca-GlyR in HEK293 cells. **h**, Membrane potential assay of coca-GlyR potency for cocaine (left; $n = 7$ separate dose–responses) and ACh (right; $n = 8$ separate dose–responses) compared with unmodified α7–GlyR ($n = 3$ separate dose–responses each). **i**, Cocaine inhibits excitability to current injection of a hippocampal neuron expressing coca-GlyR. **j**, The input resistance of hippocampal neurons expressing coca-GlyR ($n = 8$ cells) or GFP ($n = 13$ cells) in the presence of cocaine. Recovery after cocaine removal (WASH). **k**, The fold change of current that elicits an action potential (rheobase) in neurons expressing coca-GlyR ($n = 10$ (3 μM) and $n = 6$ (10 μM) cells) or GFP ($n = 4$ (3 μM) and $n = 7$ (10 μM) cells) in response to cocaine (two-way ANOVA, transgene, $F_{1,23} = 6.14$, $P = 0.02$; dose, $F_{1,23} = 0.69$; $P = 0.41$; transgene × dose, $F_{1,23} = 0.01$, $P = 0.93$). **l**, The membrane properties of neurons expressing coca-5HT3 ($n = 11$ cells), coca-GlyR ($n = 13$ cells) or GFP ($n = 14$ cells). ANOVA for capacitance, $F_{2,35} = 0.040$, $P = 0.96$; Kruskal–Wallis ANOVA by ranks for resistance: $H_2 = 1.23$, $P = 0.54$; Kruskal–Wallis ANOVA by ranks for resting potential: $H_2 = 3.58$, $P = 0.17$. Data are mean ± s.e.m. Not significant (NS), $P > 0.05$.

We performed a dose–response assessment in the coca-5HT3 and control groups after acquisition of cocaine IVSA. The dose–response curve for cocaine infusions in the control group showed a prominent peak at unit dose 0.125 mg per kg per infusion. By contrast, the coca-5HT3 rats showed suppression of the dose–response curve, consistent with a decrease in the reinforcing effects of cocaine. Notably, none of the rats reached the infusion limit (60 infusions per session at 0.5 and 1 mg per kg unit doses). A mixed-effects ANOVA showed significant effects of dose ($F_{4,94} = 29.82$, $P < 0.001$) and dose × transgene interaction ($F_{4,94} = 2.87$, $P = 0.02$). In a follow up Holm–Šidák post hoc comparison, the rats injected with AAV5-Syn::coca-5HT3-IRES-mCherry had significantly fewer infusions than the control rats at the 0.125 mg per kg ($P = 0.01$) and 0.25 mg per kg ($P = 0.04$) unit doses (Fig. 3g). Correspondingly, the rats injected with coca-5HT3 into the LHb also self-administered significantly less cocaine compared with the control rats (Fig. 3h) (mixed-effects ANOVA, dose main effect ($F_{4,94} = 57.27$, $P < 0.001$), transgene main effect ($F_{1,24} = 4.96$, $P = 0.03$), dose × transgene interaction ($F_{4,94} = 2.99$, $P = 0.02$)).

To verify the absence of effects of coca-5HT3 expression on natural behaviours in the absence of cocaine, we expressed coca-5HT3 in the LHb of different rats. Coca-5HT3 expression did not significantly affect preference for sucrose solution (two-way RM-ANOVA, transgene main effect ($F_{1,10} = 0.78$, $P = 0.39$), session main effect ($F_{1,10} = 60.8$, $P < 0.001$), transgene × session interaction ($F_{1,10} = 1.97$, $P = 0.19$)) or operant responding for sucrose pellets (two-way RM-ANOVA, transgene main effect ($F_{1,11} = 0.24$, $P = 0.63$), session main effect ($F_{5,55} = 13.64$, $P < 0.001$), transgene × session interaction ($F_{5,55} = 0.68$, $P = 0.63$)) (Extended Data Fig. 6a–d). Moreover, coca-5HT3 expression did not significantly affect the locomotor response to a novel environment (two-way RM-ANOVA, transgene main effect ($F_{1,10} = 0.04$, $P = 0.83$), session main effect ($F_{59,590} = 11.89$, $P < 0.001$), transgene × session interaction ($F_{59,590} = 0.86$, $P = 0.75$)) (Extended Data Fig. 6e,f). These experiments confirm that, in the absence of cocaine, coca-5HT3 expression in the LHb does not significantly affect locomotion, hedonic food preference or cocaine-independent operant responding for sucrose.

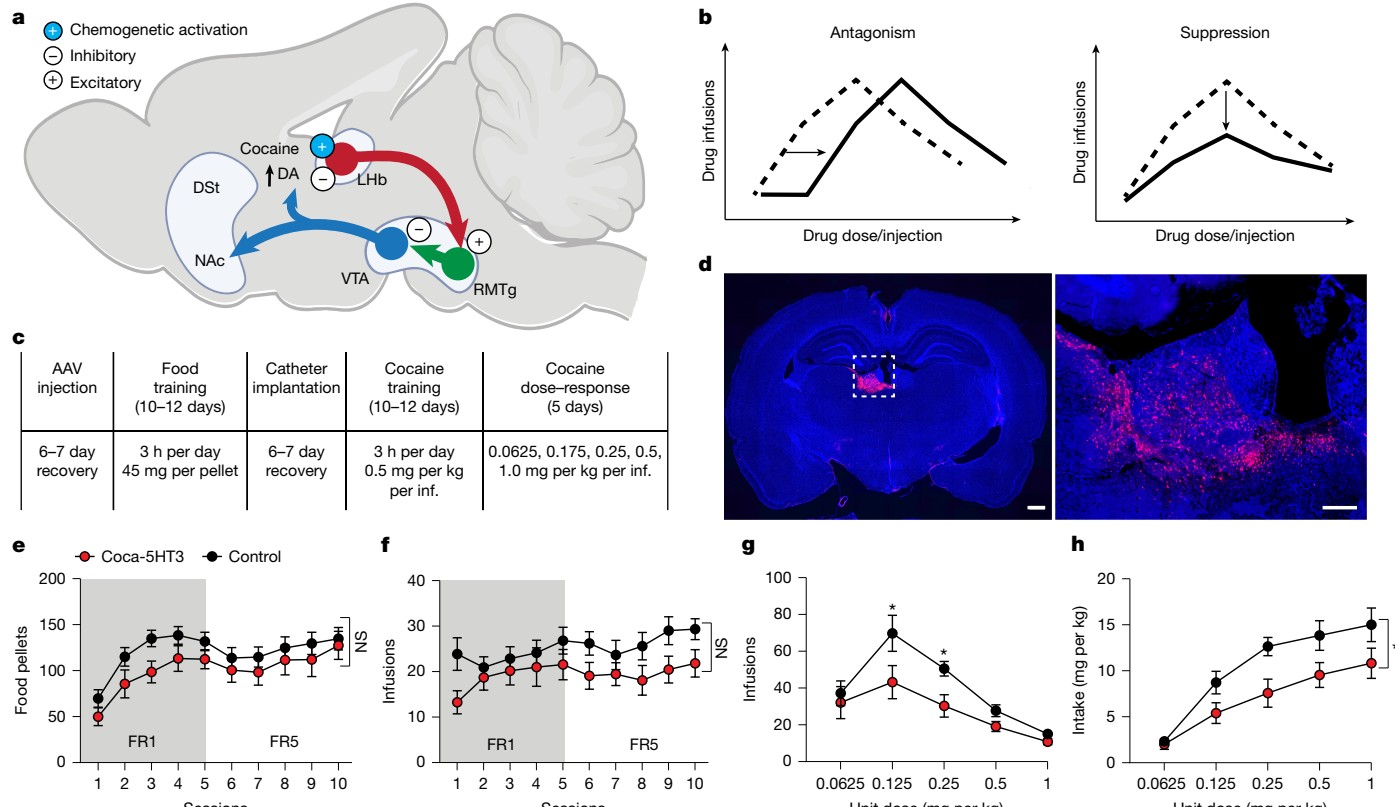

**Fig. 3 | Cocaine chemogenetics in LHb neurons suppresses cocaine-seeking.** **a**, Schematic of brain reinforcement circuits engineered to produce artificial feedback through cocaine-gated excitatory channels that oppose the natural suppression of LHb activity resulting from elevated dopamine (DA). Blue, dopamine; red, glutamate; green, GABA. DSt, dorsal striatum; NAc, nucleus accumbens. The diagram was created in BioRender. Sternson, S. (2025) https://BioRender.com/wrre5mw. **b**, Schematic of dose–response curves for drug self-administration and the potential effects of interfering with reinforcement mechanisms. **c**, Organization of the cocaine self-administration experiment. inf., infusion. **d**, Left, image of a coronal rat brain section expressing coca-5HT3-IRES-mCherry in the LHb (scale bar, 1 mm). Right, magnified image of the region indicated by the dashed box (scale bar, 200 μm). **e**,**f**, Acquisition of food (**e**; number of pellets per 3 h) and cocaine infusions per 3 h (**f**; 0.5 mg per kg per infusion) in rats with coca-5HT3 in the LHb ($n = 11$ rats) and controls ($n = 15$ rats). **g**,**h**, Rats with coca-5HT3 in the LHb ($n = 11$ rats) showed significantly lower cocaine infusions per 3 h at 0.125 mg per kg and 0.25 mg per kg (**g**; $P = 0.01$ and $P = 0.04$, respectively, Holm–Šidák post hoc comparison) and intake per 3 h at doses that maintain high-effort self-administration compared with the controls (**h**; $n = 15$ rats; mixed-effects ANOVA, dose × transgene interaction, $P = 0.02$). *$P < 0.05$. Data are mean ± s.e.m.

## Cocaine chemogenetics blunts dopamine rise

Cocaine's reinforcing and addictive properties are mediated by increases in extracellular dopamine in the nucleus accumbens[2]. Elevated LHb neuron activity negatively regulates dopamine release in the nucleus accumbens[35,40–45]. We hypothesized that cocaine-mediated activation of coca-5HT3 in the LHb would decrease cocaine-induced dopamine release in the nucleus accumbens.

To investigate this, we first used small animal positron emission tomography (PET) and the dopamine D2/D3 receptor antagonist radiotracer [18F]fallypride. In vivo [18F]fallypride binding at D2/D3 receptors is displaceable by pharmacological or task-induced elevations of extracellular dopamine and constitutes a non-invasive approach to measure brain-wide changes in dopamine release in rodents and humans[46,47]. We hypothesized that cocaine would decrease [18F]fallypride binding through displacement by elevated extracellular dopamine in control rats and that this effect would be decreased in rats injected with coca-5HT3 in the LHb (Fig. 4a,b). Using a voxel-wise statistical parametric mapping analysis, we found that cocaine significantly decreased [18F]fallypride binding in control rats in a set of voxels encompassing the medial shell and core of the nucleus accumbens and the ventromedial division of the dorsal striatum (paired $t$-test, cocaine versus saline, $t = 2.89$, $P = 0.01$) (Fig. 4c). Rats expressing coca-5HT3 in the LHb and treated with cocaine showed less [18F]fallypride displacement in the ventral striatum compared with the control rats (three-way

RM-ANOVA, time × transgene × brain region interaction, $F_{25,300} = 3.83$, $P < 0.001$) (Fig. 4c,d and Extended Data Fig. 7a). In a separate group of rats, we confirmed that expression of coca-5HT3 in the LHb did not affect the striatal dopamine concentration (unpaired $t$-test, coca-5HT3 versus control, $t = 1.07$, $P = 0.31$) (Extended Data Fig. 7b). Overall, this indicates that cocaine activation of coca-5HT3 in the LHb suppresses cocaine-induced dopamine elevation in the ventral striatum.

We used the same [18F]fallypride displacement procedure in rats expressing Cre recombinase in tyrosine hydroxylase (TH)-expressing dopamine neurons that were transduced by intracranial injection of AAV1-Syn::FLEX-coca-GlyR-IRES-mCherry to test the in vivo efficacy of coca-GlyR. *Th-cre* rats expressing coca-GlyR in VTA dopamine neurons showed less [18F]fallypride displacement in the ventral striatum compared with control rats (three-way RM-ANOVA, time × transgene × brain region interaction, $F_{12,169} = 2.44$, $P = 0.006$) (Extended Data Fig. 8) indicating that cocaine-gated inhibition of VTA dopamine neurons by coca-GlyR suppressed the typical cocaine-induced dopamine elevation in the ventral striatum.

To further investigate dopamine levels in the nucleus accumbens during cocaine chemogenetics, we performed fibre photometry in rats using a genetically encoded, fluorescent G-protein-coupled receptor (GPCR)-activation-based dopamine (GRAB_DA) sensor[48]. Rats injected with AAV5-Syn::coca-5HT3-IRES-mCherry into the LHb and controls were implanted with an optical fibre coated with AAV9-Syn::GRAB_DA1m and targeted to the shell portion of the nucleus accumbens (Fig. 4e,f).

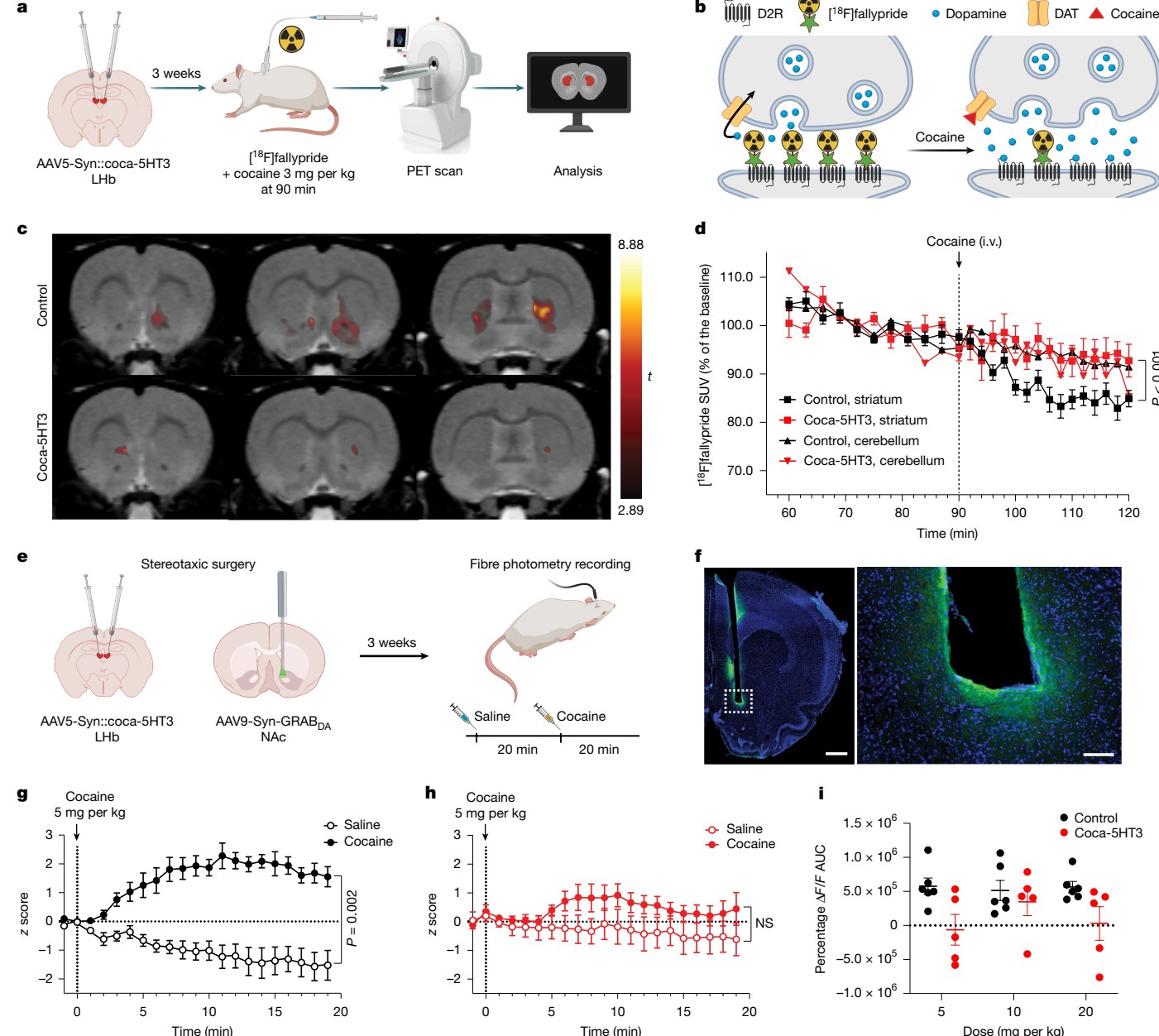

**Fig. 4 | Cocaine chemogenetics in LHb neurons suppresses cocaine-induced increases in extracellular dopamine in the NAc. a**, Schematic showing the stereotaxic delivery of AAV5-Syn::coca-5HT3-IRES-mCherry into the LHb of rats and [18F]fallypride PET procedures. The diagram was created in BioRender. Michaelides, M. (2025) https://BioRender.com/yvwycvv. **b**, Schematic showing the displacement of [18F]fallypride by cocaine detected using PET. The diagram was created in BioRender. Sternson, S. (2025) https://BioRender.com/9fog7n9. **c**, Coronal PET brain images co-registered to a magnetic resonance imaging template showing significant cocaine-induced voxel-wise decreases in [18F]fallypride binding in the ventral striatum of control rats ($n = 9$ rats) and in rats with coca-5HT3 in the LHb ($n = 5$ rats). **d**, Time–activity curves of [18F]fallypride binding (standardized uptake value (SUV)) in the ventral striatum (squares) and cerebellum (triangles) calculated as the percentage of the baseline (60–90 min) for control ($n = 9$ rats, black) and coca-5HT3 ($n = 5$ rats, red) rats, showing that cocaine selectively displaces [18F]fallypride binding (that is, increases dopamine) in the ventral striatum of control rats (three-way RM-ANOVA, time × transgene ×

brain region interaction, $F_{25,300} = 3.83$, $P < 0.001$). **e**, Schematic of the stereotaxic delivery of AAV5-Syn::coca-5HT3-IRES-mCherry and AAV9-Syn::GRAB_DA and the experimental timeline of fibre photometry procedures. The diagram was created in BioRender. Michaelides, M. (2025) https://BioRender.com/hmnai5g. **f**, Left, image of a coronal rat brain section showing the optic fibre tract and GRAB_DA expression (scale bar, 1 mm). Right, magnification of the area indicated on the left (scale bar, 400 μm). **g**,**h**, GRAB_DA response time course with saline and cocaine (5 mg per kg) injection in control rats (**g**) ($n = 5$ rats; two-way RM-ANOVA, treatment main effect, $F_{1,5} = 47.77$, $P = 0.001$; time main effect, $F_{2.2,11.1} = 1.84$, $P = 0.2$; treatment × time interaction, $F_{2.1,10.4} = 10.86$, $P = 0.002$) and rats with coca-5HT3 in the LHb (**h**) ($n = 6$ rats; two-way RM-ANOVA, treatment main effect, $F_{1,4} = 3.58$, $P = 0.13$; time main effect, $F_{1.5,6.1} = 0.73$, $P = 0.48$; treatment × time interaction, $F_{2.5,10.2} = 2.03$, $P = 0.17$). **i**, The AUC of time-course measurements with saline- and cocaine-induced (5, 10 and 20 mg per kg) GRAB_DA responses. Data are mean ± s.e.m.

Rats were anaesthetized and their optical fibre was connected to the patch cord. GRAB_DA signals were then collected for 5 min, followed by a saline intraperitoneal (i.p.) injection, and additional GRAB_DA signal

measurement for 20 min. Cocaine was then injected i.p. at either 20, 10 or 5 mg per kg, and the GRAB_DA signal was measured for an additional 20 min. Each rat was tested three times with the cocaine dose

administered in a randomized order. Compared with saline, all doses of cocaine significantly increased the $GRAB_{DA}$ response in control rats (Fig. 4g and Extended Data Fig. 9) (two-way RM-ANOVA, 5 mg per kg: treatment × time interaction ($F_{2.1,10.4} = 10.86$, $P = 0.002$); 10 mg per kg: treatment × time interaction ($F_{1.7,8.8} = 21.45$, $P = 0.005$); 20 mg per kg: treatment × time interaction ($F_{1.4,7.2} = 5.09$, $P = 0.04$)). By contrast, cocaine did not produce statistically significant increases in the $GRAB_{DA}$ response in coca-5HT3 rats at any dose (Fig. 4h and Extended Data Fig. 10) (two-way RM-ANOVA, 5 mg per kg: treatment × time interaction ($F_{2.5,10.2} = 2.04$, $P = 0.17$); 10 mg per kg: treatment × time interaction ($F_{1.1,8.8} = 1.24$, $P = 0.33$); 20 mg per kg: treatment × time interaction ($F_{1.3,5.4} = 0.37$, $P = 0.63$)). Next, we calculated the area under the curve (AUC) to test for differences in the cocaine-induced $GRAB_{DA}$ response between coca-5HT3 and control rats. Rats with coca-5HT3 had a lower $GRAB_{DA}$ response compared with control rats that was statistically significant (two-way RM-ANOVA: transgene effect ($F_{1,9} = 10.03$, $P = 0.01$); dose × transgene interaction ($F_{2,18} = 1.05$, $P = 0.37$)) (Fig. 4i). Taken together, the lack of cocaine-induced [$^{18}$F]fallypride displacement using PET and the lack of statistically significant cocaine-induced $GRAB_{DA}$ responses in coca-5HT3 rats indicate that activation of coca-5HT3 in the LHb by cocaine inhibits cocaine-induced extracellular increases in nucleus accumbens dopamine, consistent with the lower cocaine self-administration behaviour that we observed in rats expressing coca-5HT3 in the LHb.

## Discussion

Here we demonstrate a chemogenetic approach inspired by physiology that responds directly to cocaine and therefore enables the introduction of an artificial feedback process that is engaged when cocaine enters the brain and can suppress cocaine-seeking. This was implemented by engineering cocaine-specific activation of a circuit that is normally inhibited by cocaine; the activation of this circuit blunts cocaine-mediated elevation of extracellular dopamine. For this, we used protein engineering to generate ion channels that are activated by cocaine. These ion channels are highly sensitive to cocaine, and not sensitive to endogenous molecules or other addictive drugs, including nicotine, at physiologically relevant doses. Accordingly, these engineered cocaine-gated channels can be used to create artificial signalling processes opposing drug-seeking solely during drug exposure.

Chemical feedback processes are ubiquitous in biology but are challenging to study because interference with receptors involved in chemical signalling (such as dopamine receptors) typically perturbs basal functions. Cocaine chemogenetics is a biologically inspired approach to study and perturb feedback processes in organisms. By using an addictive drug as the agonist for chemogenetic neuromodulation, this approach intrinsically couples chemogenetic neuromodulation to drug self-administration and pharmacokinetic exposure in the brain. This enables a unique form of temporal control that can be used to investigate and to interfere with the drug-induced reinforcement processes of addiction.

Our findings show that activation of LHb neurons solely during cocaine exposure blunts high-effort cocaine self-administration. LHb activation did not interfere with the motor actions of cocaine-seeking or the ability of cocaine to reinforce these actions at high doses involving low effort to maintain a reinforcing dose. Instead, artificial cocaine-mediated LHb activation selectively suppressed the higher response rates necessary to maintain reinforcing exposure to cocaine at lower dose infusions. This was probably due to a reduction in the release of dopamine in the nucleus accumbens. In support of this, open-loop chemogenetic inhibition of VTA dopamine neurons similarly suppresses cocaine self-administration at intermediate infusion doses but not high doses[49].

The effect of LHb activation on cocaine self-administration is consistent with related investigations of LHb functions on reward-seeking.

Reward administration elicits phasic reduction of LHb neuron activity and omission leads to LHb activation[50]. Following the rewarding spike in cocaine after i.v. administration, brain cocaine levels quickly fall within 15 min (ref. 6) and, after this, some LHb neurons show activation[34] and cocaine also becomes aversive[36]. Cocaine activation of coca-5HT3 in the LHb artificially induces the anti-rewarding effect of LHb activation during the cocaine-exposure period, therefore opposing high-effort cocaine self-administration. This perturbation is intrinsically linked to the rise and fall of the cocaine concentration. This is a consequence of the rapid activation of coca-5HT3 by cocaine and the rapid deactivation of coca-5HT3 after cocaine removal. The specific LHb subpopulations that are modulated by cocaine are not established; therefore, further investigation is needed to precisely target cocaine chemogenetics as a negative-feedback control system for cocaine reinforcement. The behavioural effect of reduced cocaine self-administration is consistent with other studies showing that LHb activation reduces reward-seeking behaviours[51,52] and is associated with depression[53]. Here cocaine-induced activation of the LHb blunted the cocaine-induced dopamine rise without suppressing dopamine below the baseline levels, which is desirable to counteract the reinforcing effects of cocaine without gross disruptions to other functions of dopamine signalling. The relationship between motor vigour and elevated dopamine signalling[54–56] probably contributes to the large effects of this chemogenetic process to suppress high effort responding for cocaine. One limitation of this study is that it focused solely on male rats and included the use of two different rat strains. Further investigation is needed to examine the extent to which these effects generalize to female rats and other rat strains or species.

Cocaine chemogenetics could be further deployed to investigate the contributions to addiction behaviours by other neural populations. For example, cocaine-induced chemogenetic activation of a pain- or sickness-inducing neuron population may also alter drug self-administration, which would be analogous to the mechanism of the alcohol-use-disorder treatment drug disulfiram. Importantly, cocaine chemogenetics is a means to selectively perturb the outcome evaluation phase after drug ingestion, which is challenging even with rapid optogenetic methods because these perturbations are difficult to precisely yoke to drug exposure in the brain. In the future, chemogenetic receptors can be developed with selectivity for additional addictive drugs or even drug combinations, which would facilitate efforts to study their neural circuit mechanisms using drug-mediated chemogenetic neuromodulation. Based on what is learned with addictive drugs, these closed-loop chemogenetic approaches may be further applied to other body–brain signalling processes, such as those involving hormones or nutrients.

The chemogenetic receptors reported here are specific for cocaine over food rewards. This is important because cocaine alters signalling pathways that are also used for processing natural rewards. Indeed, activation of the LHb suppresses food consumption[51,52]; it is therefore desirable to activate the LHb only during cocaine taking. The shared role of drug-seeking circuits with motivation for healthy physiologically relevant rewards has been a major impediment to developing pharmacotherapies for addictive drugs. We found that cocaine-gated chemogenetic LHb activation suppresses cocaine intake in rats using a well-validated preclinical model of cocaine abuse liability. As these chemogenetic ion channels are specific for cocaine, they may eventually lead to gene therapies selective for cocaine addiction without affecting enjoyment of natural rewards.

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

## Methods

### DNA constructs

The α7–5HT3 and α7–GlyR sequences were the same as reported previously[11]. Site-directed mutagenesis was performed using QuickChange Lightning SDM (Agilent). For these constructs, the α7 nAChR LBD was based on the sequence of human *CHRNA7*. The 5HT3 IPD was based on the mouse sequence. The GlyR IPD was based on the human sequence. For testing in human embryonic kidney (HEK) cells, channels were expressed from pcDNA3.1, (Invitrogen/Thermo Fisher Scientific).

For Cre-dependent rAAV, constructs containing Coca-GlyR followed by IRES-mCherry were cloned into an AAV2-Synapsin(Syn)-FLEX vector[57] in the reverse orientation with respect to the *Syn* promoter. Using these constructs, rAAV serotype 1 was produced by the Janelia Molecular Biology Core Facility.

Coca-5HT3 (α7(L141G,G175K,Y210F,Y217F)–5HT3) or coca-GlyR (α7(L141G,G175K,Y217F)–GlyR) followed by IRES-mCherry were cloned into an AAV2 vector downstream of the *Syn* promoter and were also packaged as AAV serotype 5 (Janelia).

For electroporation and in vitro electrophysiology in CA1 pyramidal cells, mutant channel cDNAs were cloned into a pCAG expression vector with an upstream CAG promoter and a downstream IRES-GFP sequence.

The numbering scheme used for ion channel amino acid residue positions is based on the unprocessed pre-protein sequence derived from the cDNA sequences.

### Structural model of cocaine–LBD interactions

The crystal structure of cocaine in a complex with AChBP was obtained from the Protein Data Bank (PDB: 2PGZ). Amino acid substitutions were made in PyMol for residues near cocaine in the crystal structure for the homologous residues in the human protein sequence for display purposes. The model was used to guide exploration of residues to be mutated for effects on cocaine agonism.

### Ion-channel testing using a membrane potential assay

ATCC CRL-1573 cells (HEK293, passages 40–49) were plated into Corning CellBIND plates (10 cm diameter), at $3 \times 10^6$ cells per plate and grown for 18–24 h to achieve about 80% confluency. Although the HEK293 cell line is frequently misidentified, we used obtained it directly from the ATCC and used these cells because the chimeric ion channels constructs are efficiently transfected and express well in this cell line. Cell lines were tested for mycoplasma contamination. If contaminated cultures were detected, then they were not used for experiments.

To transfect cells, Opti-MEM (500 µl, Invitrogen), DNA plasmid (10 µg) and FuGene HD (30 µl, Roche) were combined, incubated for 15 min at room temperature and the mixture was then added dropwise to cells in a 10 cm² plate. Cells were incubated at 37 °C in 8% $CO_2$/air. After an additional 18–20 h, the medium was aspirated and cells were rinsed (1× PBS, 5 ml), treated with 0.05% trypsin-EDTA (3.0 ml, 5 min) and neutralized with DMEM + 10%FBS (7.0 ml). Cells were pooled, centrifuged (100*g*, 2 min) and resuspended in DMEM + 10% FBS then counted (Beckman Coulter Vi-Cell XR cell analyzer). Cells were then plated in 96 well BIOCOAT poly-D-lysine black/clear cellware with a cell density of $6 \times 10^4$ live cells in 200 µl per well. Cells were incubated another 22–26 h (37 °C in 8% $CO_2$) then medium aspirated, and the FLIPR membrane potential assay solution was added and incubated at room temperature for approximately 20 min.

Membrane potential assay solution was prepared with a bottle of Molecular Devices membrane potential assay kit–Blue (R8034) dissolved in 200 ml of HEPES-buffered Earle's balanced salt solution (HEBSS). Drug compound plates were prepared in Nalge Nunc 96-well conical-bottom plates in HEBSS ( < 0.5% DMSO) and tested at concentration intervals from 0.1 µM to 100 µM. (<0.1% DMSO)

Amphetamine and cocaine (Sigma-Aldrich) were dissolved in saline to a 100 mM stock before making the dilution series. Nicotine was dissolved to 100 mM stock in saline from >99% liquid (Sigma-Aldrich). All of the other controlled substances were isolated in 1 mg quantities using the SpeedVac (Thermo Fisher Scientific) from Cerilliant ACS Chemical Standard solutions in methanol (Sigma-Aldrich) and dissolved in DMSO stock to 100 mM. DMSO alone was also tested to ensure that there was no response at these dilutions. Endogenous amines for testing at coca-5HT3 (Extended Data Fig. 3) were dissolved in saline to 100 mM stocks.

To measure ion-channel activation, we used the Hamamatsu FDSS 6000 plate reader and liquid handling system. The assay plates were scanned at 1 Hz (excitation, 472/30 nm; emission, 540/40 nm). Compound solutions (50 µl) were simultaneously delivered from a 96-well compound plate after 10 baseline scans, which were followed by 170 additional scans. To calculate the MP $EC_{50}$, the maximum response for each well was extracted, normalized to the maximum response for each compound and sigmoidal dose–response curves were calculated using software written in MATLAB (MathWorks Central: ec50.m, C. Evangelista, v.1.0, 14 January 2004) and the nlinfit function from the Statistics Toolbox.

### HEK293 cell electrophysiology

Channel mutants and agonists were further characterized in whole-cell voltage-clamp experiments performed on HEK293 cells. At passage 44–48, cells were plated at low density ($1 \times 10^3$ per cm²) on poly-D-lysine-coated glass coverslips, then transfected in 10 cm² wells using the Fugene 6 transfection reagent (Roche) with DNA plasmids containing both an ion channel cDNA (pcDNA3.1, 2 µg per well) and cDNA for GFP (pCMV, 0.1 µg per well), followed by replacement of the medium after 3 h. Recordings were made 24–72 h after transfection.

Electrophysiological recordings were performed in whole-cell voltage-clamp mode, on an inverted fluorescence microscope (Olympus IX-51) using pClamp software and a Multiclamp 200B amplifier (Molecular Devices). Electrophysiological data was filtered at 4 kHz, digitized at 10 kHz using a Digidata 1400 interface (Molecular Devices), recorded on a PC (Dell) and analysed using custom MATLAB software (MathWorks). Solutions were delivered from a gravity fed perfusion system through a capillary tube placed approximately 100 mm from the recorded cell using an eight-way manifold pinch valve system (Nanion), which was controlled from the acquisition software. This allowed for up to seven concentrations of agonist to be tested. The duration of agonist application was based on pilot experiments to determine the time required to reach a steady-state current response for the channel. External solution (HEBSS) contained 140 mM NaCl, 10 mM HEPES, 5 mM glucose, 4 mM KCl, 2 mM $CaCl_2$ and 1 mM $MgCl_2$. The whole-cell pipette internal solution contained 87 mM CsGluconate, 31 mM CsCl, 5 mM NaCl, 10 mM HEPES, 10 mM EGTA and 3 mM MgATP. Osmolarities were within 285–305 mOsm l$^{-1}$ and the pH was adjusted to 7.3–7.4 with NaOH.

### Electrophysiology of cultured hippocampal neurons and LHb acute brain slices

Experiments and procedures complied with ethical regulations for animal testing and research and were approved by the Janelia Research Campus and UCSD animal care and use committees. Hippocampal neuron cultures were prepared by modification of a previously reported method[58]. Hippocampi were dissected from postnatal day 0 neonatal rat pups (Charles River) in HEPES buffered Hank's solution containing 100 U ml$^{-1}$ penicillin and 10 µg ml$^{-1}$ streptomycin. Tissue was dissociated in papain (Worthington, PAP2) in dissection buffer for 30 min at 37 °C. After digestion, papain was removed and tissue was dissociated in plating medium (MEM plus 10% fetal bovine serum, 28 mM glucose, 2.4 mM NaHCO₃, 100 µg ml$^{-1}$ transferrin, 25 µg ml$^{-1}$ insulin, 2 mM L-glutamine, 100 U ml$^{-1}$ penicillin and 10 µg ml$^{-1}$ streptomycin). The cell suspension was passed through a 70 µm filter and centrifuged at 90*g* for 7 min. The resulting cell pellet was resuspended in plating medium, counted and assayed for cell viability.

pCAG::coca-GlyR-IRES-GFP or pCAG::coca-5HT3-IRES-GFP plasmids were transfected by electroporation using the Lonza Nucleofector system (P3 Primary Cell 96-well Kit, Lonza) then cells were plated onto poly-D-lysine-coated 13 mm dia #1 glass coverslips in the wells of a 24-well plate at 50,000 cells per well. Each well received 60 µl of cell suspension plus 60 µl of NbActiv4 medium (BrainBits) and was incubated at 37 C in 5% $CO_2$ for 4 h for initial cell attachment. NbActiv4 medium (1 ml) was then added to each well and cultures kept for the duration of the experiment. Weekly medium changes were made by replacing 0.5 ml medium from each well with fresh NbActiv4.

Whole-cell recording internal solution contained 135 mM potassium gluconate, 6.7 mM KCl, 10 mM HEPES, 1.0 mM EGTA, 4.0 mM Na2ATP and 0.3 mM Na3GTP. The osmolarity was 285–305 mOsm $l^{-1}$ and the pH was within 7.3–7.4. Whole-cell recording pipettes had tip resistance between 1.5 and 4 MΩ and whole-cell series resistance was 6–29 MΩ (mean ± s.e.m., 17.2 ± 0.6 MΩ). The external solution contained: 140 mM NaCl, 10 mM HEPES, 5 mM glucose, 4 mM KCl, 2 mM $CaCl_2$ and 1 mM $MgCl_2$. Tight seals (>4 GΩ) were made in voltage-clamp mode and, immediately after patch disruption to establish the whole-cell recording configuration, the resting membrane potential was measured by switching briefly to the 'I = 0' mode on the Multiclamp Commander (Molecular Devices). After switching back to voltage-clamp mode, the other cell membrane properties were measured using the automated cell capacitance and membrane resistance compensation circuitry on the same device. Series resistance compensation was not applied. Membrane properties during current-clamp recordings were monitored by small current injections (−20 pA, 200 ms, 1 Hz). Thresholds for activation were estimated in current-clamp mode by injection of sequentially increasing depolarizing current steps (either 5 pA or 50 pA steps depending on whether a large shunt was observed due to chemogenetic receptor activation by cocaine). The threshold was measured as the amplitude of current injection that induced the first action potential.

For analysis of coca-5HT3 effects on the passive membrane properties of LHb neurons, C57BL/6J mice (male, aged 4–5 weeks) were injected with AAV5-Syn::coca-5HT3-IRES-mCherry (titre, $4 \times 10^{12}$ viral genomes (vg) per ml) or with a control virus AAV5-Syn::IRES-mCherry (titre: $2 \times 10^{12}$ vg per ml) in the LHb (anteroposterior (AP): −1.55 mm relative to bregma; mediolateral (ML): +0.45 mm; dorsoventral (DV): −2.65 mm, −2.75 mm, −2.85 mm, 100 nl at each depth). After 2–5 weeks, brain slices were prepared. In brief, animals were deeply anaesthetized with isoflurane, decapitated and the brains removed in ice-cold sectioning solution (described below). The brains were then mounted onto a stage using Krazy Glue, and coronal brain slices (250 µm) were cut using a cooled tissue slicer (Leica 1200S). Slices were made in an ice-cold, low-sodium, low-calcium solution containing: 135 mM N-methyl-D-glucamine, 2.2 mM KCl, 20 mM $NaHCO_3$, 1.2 mM $NaH_2PO_4$, 10 mM glucose, 0.25 mM $CaCl_2$, 1.75 mM $MgCl_2$ and 113 mM HCl (HCl 25% solution, 6.85 M, 16.5 ml $l^{-1}$) then kept in recording saline, with 1 mM ascorbic acid and 2 mM pyruvic acid added, until needed (30 min to 5 h). N-Methyl-D-glucamine was used as a sodium replacement. Slices were held down in a glass-bottomed chamber, perfused and viewed on the stage of an upright Olympus BX-51 fluorescence microscope using infrared gradient contrast optics. Transfected neurons were identified by their mCherry fluorescence. Electrophysiological recordings were made in whole-cell current-clamp mode, using a Multiclamp 200B amplifier (Molecular Devices) and MATLAB Ephus software for acquisition[59]. Data were filtered at 4 kHz, digitized at 10 kHz using the National Instruments interface hardware and recorded on a PC.

### Radioligand-binding assays in HEK293 cells

HEK293 cells were transfected with 5 µg per dish of plasmids encoding coca-5HT3 or coca-GlyR (in pcDNA3.1) and collected 48 h after transfection. Cells were suspended in Tris-HCl 50 mM pH 7.4 supplemented with protease inhibitor cocktail (1:100, Sigma-Aldrich).

HEK293 cells were disrupted with a 6 Polytron homogenizer (Kinematica). Homogenates were centrifuged at 48,000$g$ (50 min, 4 °C) and washed twice in the same conditions to isolate the membrane fraction. Protein was quantified using the bicinchoninic acid method (Pierce). Membrane suspensions (50 µg of protein per ml) were incubated in 50 mM Tris-HCl (pH 7.4) containing 8 mM $CaCl_2$, 2 nM of [$^3$H]ASEM (26 Ci $mmol^{-1}$, Novandi Chemistry) and increasing concentrations of the tested compounds during 2 h at room temperature as previously described[12]. Non-specific binding was determined in the presence of 100 µM of non-radiollabeled ASEM. Free and membrane-bound radioligands were separated by rapid filtration of 500 µl aliquots in a 96-well plate harvester (Brandel) and washed with 2 ml of ice-cold Tris-HCl buffer. Microscint-20 scintillation liquid (65 µl per well, PerkinElmer) was added to the filter plates, plates were incubated overnight at room temperature and radioactivity counts were determined in the MicroBeta2 plate counter (PerkinElmer) with an efficiency of 41%. Two-site competition curves were fitted using Prism 10 (GraphPad). $K_i$ values were calculated using the Cheng–Prusoff equation.

### Animals for behavioural experiments

Experiments and procedures complied with ethical regulations for animal testing and research, followed the NIH guidelines and were approved by the NIDA animal care and use committee. The experimenters were blinded to the group allocation. Rats were single housed on a reverse light cycle (lights off 07:00; lights on 19:00). Sample sizes were estimated using G*Power or based on experience from past work.

### Open-field locomotion

Adult male Sprague–Dawley rats were bilaterally injected (0.5 µl per side) with AAV5-Syn::coca-5HT3-IRES-mCherry in the LHb (10° angle, AP relative to bregma: −3.8 mm; ML, ±1.7 mm; DV, −5.2 mm) ($n = 6$) or exposed to sham surgery ($n = 6$). Then, 4 weeks later, the rats were assessed for open-field locomotion. Each rat was placed in a clear Plexiglas box (40 cm × 40 cm × 30 cm) and activity was recorded by infrared beam brakes for 60 min (Opto-varimex ATM3, Columbus Instruments). The distance travelled was plotted as mean ± s.e.m. centimetres between the control and coca-5HT3 rats.

### Sucrose preference

Rats from the open-field locomotor experiments were presented with two bottles of water in their home cage to acclimatize them to the procedure for 1 day. The next day, one bottle was replaced with a 1% sucrose solution dissolved in water. After 24 h, the consumed volume was noted in ml and the position of the bottles was switched to control for potential side preference. On day 3 the consumed volume was calculated for both water and sucrose 24 h after position switch. This volume was then plotted as percentage mean ± s.e.m. consumption between control and coca-5HT3 rats.

### Sucrose pellet operant responding

Adult male Sprague–Dawley rats were bilaterally injected (0.5 µl per side) with AAV5-Syn::coca-5HT3-IRES-mCherry in the LHb (10° angle; AP relative to bregma, −3.8 mm; ML, ±1.7 mm; DV, −5.2 mm) ($n = 8$) or exposed to sham surgery ($n = 5$). Then, 4 weeks later, the rats were restricted to 18–20 g of chow per day and trained to press a lever for 20 mg sucrose pellets on an FR1 schedule (1 h per day for 7 days). Only the active lever was extended during the session. Presses on this lever activated the food magazine to dispense one pellet (Dustless Precision Pellets, F07595, 20 mg, Sucrose; Bio-Serv) accompanied by a 20 s cue light and timeout. Responses during the timeout period were recorded but did not result in pellets being dispensed.

### Dopamine levels

Rats from the sucrose preference experiment were bilaterally injected (0.5 µl per side) with AAV5-Syn::coca-5HT3-IRES-mCherry in the LHb

(10° angle; AP relative to bregma, −3.8 mm; ML, ±1.7 mm; DV, −5.2 mm) ($n = 6$) or exposed to sham surgery ($n = 6$). Then, 6 weeks later, the rats were euthanized, and brain tissue was extracted and flash-frozen in isopentane. The tissue was then chunked, removing the olfactory bulbs and coronally sectioning at the optic chiasm. The tissue that contained the bilateral striatum was then homogenized and analysed for dopamine levels using an enzyme-linked immunosorbent assay (ELISA; Abcam, ab285238). Dopamine levels were plotted as the mean ± s.e.m. pg per mg between control and coca-5HT3 rats.

## Cocaine IVSA

Male Long Evans rats were anaesthetized with isoflurane (1.5–2%). The rats were then placed onto a stereotaxic frame (Kopf) and injected with the AAV vector AAV5-Syn::coca-5HT3-IRES-mCherry (0.5 µl, $4.95 × 10^{12}$ vg per ml) into the LHb (AP relative to bregma, −3.8 mm; ML, ±0.7 mm; DV, −5.25 mm, at a 10° angle) ($n = 11$) or underwent sham surgery ($n = 15$).

Approximately 1 week later, the rats were trained to self-administer food in daily operant sessions. For food training, rats were restricted to four large chow pellets per day and the body weight was monitored daily. Rats were tested for 10 days, and the sessions lasted for 3 h. For the first 5 days, the rats were tested using a fixed ratio 1 (FR1) schedule of reinforcement and were then switched to an FR5 schedule for the 5 subsequent days. Each lever press led to delivery of a 45 mg food pellet (maximum of 200 pellets per session) and activation of a cue light above the lever for 1 s. Each reinforced lever press was followed by a 20 s timeout during which lever responses were recorded but had no programmed consequences.

After these operant sessions, the rats underwent surgery for jugular vein catheterization. The rats were anaesthetized with isoflurane (1.5–2%). Catheters made from Silastic tubing attached to a modified 22-gauge cannula (Plastics One, C313G-5up) and cemented to polypropylene mesh (Elko Filtering, 05–1000/45) were inserted into the jugular vein, and the mesh was fixed to the mid-scapular region of the rat. Rats were injected with carprofen (2.5 mg per kg, subcutaneous, Norbrook) after surgery and on the next day. Rats recovered for 6 days before self-administration procedures. Catheters were flushed daily with gentamicin (4.25 mg ml$^{-1}$, Fresenius Kabi, 1002) dissolved in sterile saline. If catheter failure was suspected, catheter patency was tested with a short-acting barbiturate anaesthetic Brevital (methohexital sodium, 10 mg ml$^{-1}$ in buffered saline, 0.1–0.2 ml injection volume, i.v.).

After approximately 1 week, the rats began daily cocaine IVSA operant training sessions. Rats were tested using an FR1 schedule of reinforcement and were then switched to an FR5 schedule. In these sessions, a response on the lever previously paired with food had no programmed consequences (that is, inactive lever). A response on the cocaine-paired lever (that is, active lever) led to the infusion of a 0.5 mg per kg dose of cocaine (maximum of 60 infusions per session) and coincided with activation of a cue light located above the active lever for 20 s (that is, timeout period). During this time, lever responses were recorded but had no programmed consequences. After cocaine training, the rats were tested for IVSA using different cocaine unit doses (0.0625, 0.125, 0.25, 0.5 and 1 mg per kg per infusion). The procedures were identical to the initial sessions. The order of unit dose testing was randomized across rats. Rats could commit a maximum of 60 infusions when tested at the 0.5 and 1 mg per kg unit doses but no infusion limit was present at other unit doses. Statistical analyses were performed using Prism 10 (GraphPad).

## PET analysis with [18F]fallypride

Male Sprague–Dawley rats were anaesthetized with isoflurane (1.5–2%) and then placed onto a stereotaxic frame (Kopf) and injected with AAV5-Syn::coca-5HT3-IRES-mCherry (0.5 µl, $4.95 × 10^{12}$ vg per ml) into the LHb (AP relative to bregma, −3.8 mm; ML, ±0.7 mm; DV, −5.25 mm, at a 10° angle) ($n = 5$) or underwent sham surgery ($n = 9$).

Male *Th-cre* rats were anaesthetized with isoflurane (1.5–2%) and then placed onto a stereotaxic frame (Kopf) and injected with AAV1-Syn::FLEX-coca-GlyR-IRES-mCherry (0.5 µl, $6 × 10^{12}$ vg per ml) into the VTA (AP, −6.0 mm relative to bregma; ML, ±1.9 mm; DV, −8.3 mm at a 10° angle) ($n = 9$) or underwent sham surgery ($n = 6$). Approximately 3 weeks later, the rats were anaesthetized, placed inside the PET scanner (Mediso NanoScan PET/CT), injected with [18F]fallypride (-13 MBq) and then scanned using PET. [18F]fallypride reaches steady-state by around 45 min after its injection and then remains stable for at least 150 min[60]. We therefore defined the baseline [18F]fallypride binding from 60 to 90 min after its injection and then, to elicit an increase in extracellular dopamine, at 90 min after [18F]fallypride injection, we injected rats with cocaine (3 mg per kg, i.v.) and continued to scan for another 30 min (90–120 min). The PET data were reconstructed and corrected for dead-time and radioactive decay and then co-registered to an magnetic resonance imaging template as previously described[46,60,61]. Co-registered images were analysed using one-way ANOVA and the resulting parametric images were filtered for statistically significant ($P < 0.01$) clusters larger than 100 contiguous voxels. All statistical parametric mapping (SPM) analyses were performed using MATLAB R2016 (MathWorks) and SPM12 (University College London)[62]. Next, additional quantitative assessments of PET images were performed using the PMOD software environment (PMOD Technologies). We generated time–activity curves using the co-registered dynamic PET images and the significant voxel clusters defined by the voxel-wise SPM analysis as the volume of interest. The standardized uptake value (SUV) was calculated using the equation SUV = $C$/(dose/BW) where $C$ is the tissue concentration of [18F]fallypride (kBq cm$^{-3}$), dose is the administered dose (MBq) and BW (kg) is the animal's body weight. Statistical analyses were performed using Prism 10 (GraphPad).

## Fibre photometry of dopamine release after cocaine administration

Male Sprague–Dawley rats were anaesthetized with isoflurane (1.5–2%) and then placed onto a stereotaxic frame (Kopf) and injected with AAV5-Syn::coca-5HT3-IRES-mCherry (0.5 µl, $4.95 × 10^{12}$ vg per ml) into the LHb (AP, −3.8 mm relative to bregma; ML, ±0.7 mm; DV, −5.25 mm, at a 10° angle) ($n = 5$) or underwent sham surgery ($n = 6$). Rats were also implanted with an optic fibre (400 µm core, 0.50 NA) attached to a 2.5 mm diameter ferrule (Thorlabs) directed to the medial shell of the nucleus accumbens (AP, +1.7 mm relative to bregma; ML, ±0.9 mm; DV, −7.0 mm). To ensure that AAV9-Syn::GRAB$_{DA1m}$ was transduced at the direct vicinity of the fibre tip, we coated the fibre with the AAV (0.4 µl, $5 × 10^{12}$ vg per ml) using a silk fibroin film before its implantation as previously described[63].

For fibre photometry, dopamine (GRAB$_{DA}$) dynamics were measured using a Tucker-Davis Technologies (TDT) system in anaesthetized rats. The dopamine and isosbestic signals were excited by 465 (-40–50 µW) and 405 nm (-20 µW) LEDs, respectively. Data were acquired at 1 kHz. On the testing day, rats were anaesthetized (isoflurane: induction, 5%; maintenance, 3%) and attached to a photometry patch cord (400 µm core, 0.50 NA). GRAB$_{DA}$ signals were then collected for 5 min, followed by a saline i.p. injection, and additional GRAB$_{DA}$ signal measurement for 20 min. Cocaine was then injected i.p. at 20, 10 or 5 mg per kg, and the GRAB$_{DA}$ signal was measured for an additional 20 min. Each rat was tested three times with cocaine doses administered in randomized order. Fibre photometry analyses were performed using scripts in MATLAB R2023b (MathWorks) based on codes from the Barker (https://github.com/djamesbarker)[64] and Lerner (https://github.com/talialerner/)[65] laboratories. The dopamine signal was fitted with the isosbestic signal [$\Delta F/F$] and transformed to a $z$ score for comparison of the dopamine signal between rats[64]. The GRAB$_{DA}$ signal was normalized to values at 1 min before saline or cocaine injection and shown as the average $z$-scored activity per min. The AUC of the percentage

$\Delta F/F$ of GRAB$_{DA}$ fluorescence was calculated in a 20 min bin after the saline or cocaine injection. Fibre photometry data are represented as mean ± s.e.m. Statistical analyses were performed using Prism 10 (GraphPad).

## Immunohistochemistry

Rats were anaesthetized with isoflurane and transcardially perfused with PBS, followed by 4% PFA. Brains were stored overnight in 4% PFA at 4 °C and then transferred to PBS with 30% sucrose for 3 to 4 days at 4 °C. Brains were frozen and sectioned on a cryostat (30 μm to 50 μm; Leica) and slices were collected in PBS. On the first day, the sections were blocked with 0.1% saponin and 1% BSA in physiological saline for 30 min at room temperature and incubated with specific antibodies: rat monoclonal anti-mCherry (1:1,000 diluted in 0.01% saponin and 0.1% BSA in physiological saline, Invitrogen, M11217) or chicken polyclonal anti-GFP antibody (1:2,000 diluted in 0.01% saponin and 1% BSA in physiological saline, Abcam, ab13970) at 4 °C overnight. The slices were soaked in physiological saline (three times for 5 min at room temperature), then incubated with a secondary antibody in 0.01% saponin and 0.1% BSA dissolved in physiological saline: goat-anti rat immunoglobulin G (IgG) (H+L) cross-absorbed, Alexa Fluor 594 (1:500, Invitrogen, A11007), or goat-anti chicken immunoglobulin Y (IgY) H&L (Alexa Fluor 488) (1:1,000, Abcam, ab150173) for 2 h at room temperature (protected from light). After secondary antibody incubation, the slices were washed with physiological saline (three 5 min washes) and then mounted onto SuperFrost Plus Slides, counterstained with Fluoromount-G medium (Southern Biotech) and coverslipped. Images were taken using a Leica LMD6 microscope equipped with the Leica DFC7000T camera and processed using LasX software.

## Statistics

Statistical analysis was performed using SigmaStat or GraphPad Prism. We used mixed, two-way or three-way ANOVA taking repeated measures into account when appropriate or independent $t$-tests or paired-sample $t$-tests, as defined in the text. Statistical tests were two-sided. Post hoc tests were performed using Holm–Šidák multiple-comparison correction. Data are shown as mean ± s.e.m.

## Reporting summary

Further information on research design is available in the Nature Portfolio Reporting Summary linked to this article.

## Data availability

All data supporting the in vivo findings are available within the Article. DNA constructs will be available at Addgene (https://www.addgene.org/Scott_Sternson/) under a materials transfer agreement. Source data are provided with this paper.

## Code availability

Analysis code is available at GitHub and Zenodo[66] (https://github.com/Osolca/F-Phot-Analysis and https://doi.org/10.5281/zenodo.15665589).

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

**Acknowledgements** We thank the staff at the Janelia Research Campus core facility staff (M. Ramirez performed mutagenesis and subcloning for this research) for support; M. Hiramoto for performing some electrophysiology characterization; and the staff at the National Institute on Drug Abuse Translational & Analytical Core (S. Jackson and L. Kryszak) and the National Institute on Drug Abuse Genetic Engineering and Viral Vector Core (C. Richie) for support. This research was supported by HHMI (S.M.S. and C.J.M.); the W.M. Keck Foundation (S.M.S.); and National Institute on Drug Abuse Intramural Research Program ZIA DA000069 (M.M.). This Article is subject to HHMI's Open Access to Publications policy. HHMI laboratory heads have previously granted a non-exclusive CC BY 4.0 license to the public and a sublicensable license to HHMI in their research articles. Pursuant to those licenses, the author-accepted manuscript of this Article can be made freely available under a CC BY 4.0 license immediately on publication.

**Author contributions** Conceptualization: S.M.S. Methodology: S.M.S., C.J.M., J.L.G., J.B., O.S., M.R.L. and M.M. Investigation: J.L.G., C.J.M., J.B., O.S., F.P.C., M.R.L., R.C.B., M.L.C., E.N.V., S.L., L.W., I.S. and W.D. Funding acquisition: S.M.S. and M.M. Project administration: S.M.S. and M.M. Supervision: S.M.S. and M.M. Writing—original draft: S.M.S. and M.M. Writing—review and editing: S.M.S., M.M., J.L.G., C.J.M., J.B., O.S., F.P.C., M.R.L., R.C.B., M.L.C., E.N.V. and S.L.

**Competing interests** S.M.S. and C.J.M. have a pending patent application on this technology (US patent application 20240002463-A1) as well as additional issued patents on related chemogenetic technologies. S.M.S. is a consultant for Kriya Therapeutics, which is focused on therapeutic applications of chemogenetics. M.M. is a principal investigator on a cooperative research and development agreement between NIDA and Kriya Therapeutics. The other authors declare no competing interests.

**Additional information**
**Correspondence and requests for materials** should be addressed to Michael Michaelides or Scott M. Sternson.

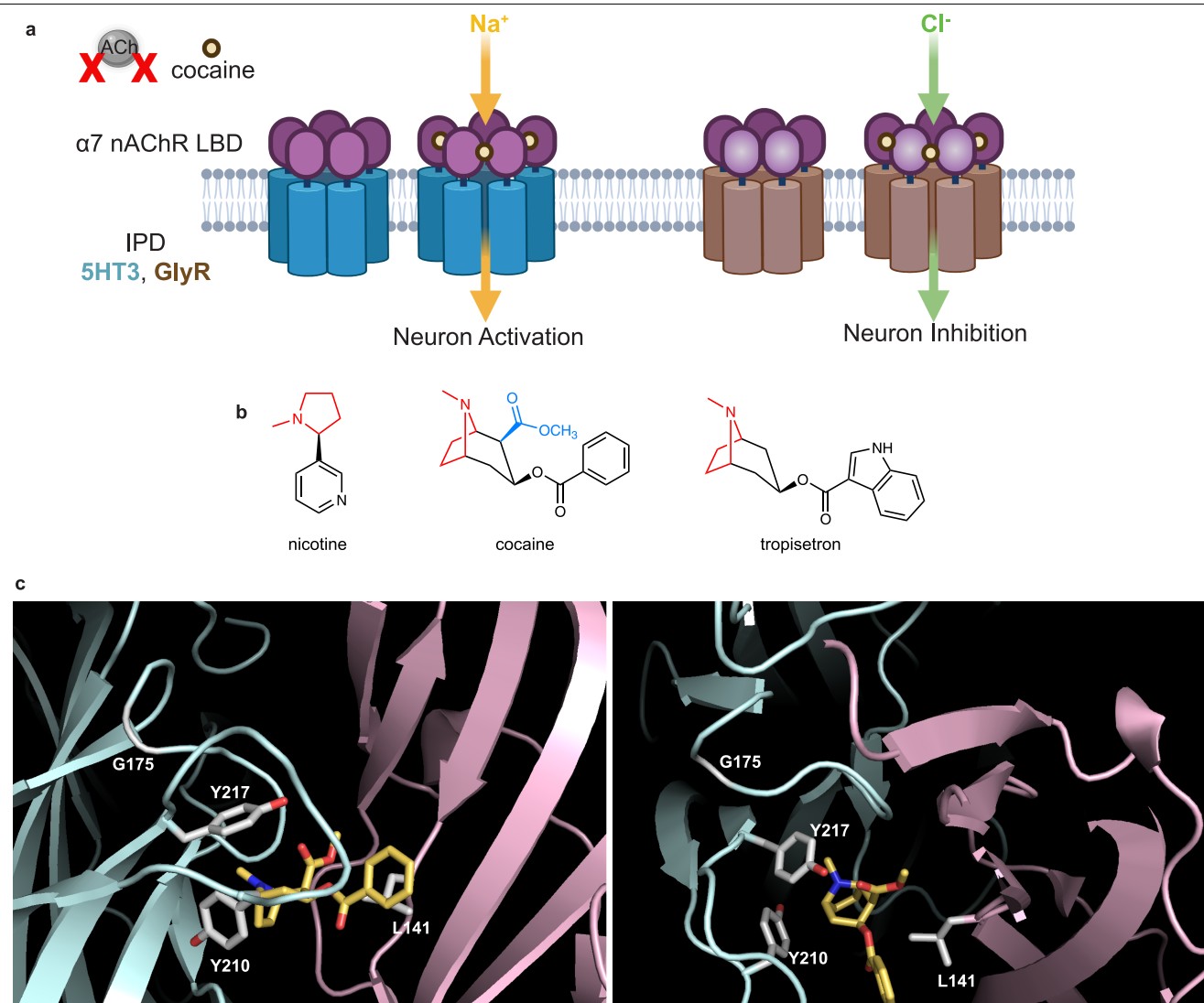

**Extended Data Fig. 1 | Chimeric ion channels that are activated by cocaine. a**, Chimeric ion channels were created from the ligand binding domain (LBD) of the α7 nAChR that was spliced to either the IPD of 5HT3 or GlyR and produced channels for neuron activation or inhibition, respectively. Mutated chimeric channel subunits homopentamerize to form LGICs. Mutations in the LBD conferred novel cocaine agonism and reduced ACh sensitivity. Created in BioRender. Sternson, S. (2025) https://BioRender.com/2uq2ikb. **b**, Structural elements that correspond to the nicotine pharmacophore are highlighted in red. Acetoxymethyl ester portion of cocaine that is proposed to adversely affect cocaine agonism of the unmodified α7 nAChR is highlighted in blue. Structural relationship to tropisetron, an α7 nAChR agonist, is shown on the right. **c**, Cocaine binding to AChBP (modified from PDB: 2pgz). Two views of cocaine (yellow) bound to AChBP at the interface of two protomer subunits (cyan & pink) showing 4 amino acid side chains in grey that are from homologous α7 nAChR human sequence and are mutated in coca-5HT3. Homologous α7 nAChR pre-protein numbering is shown.

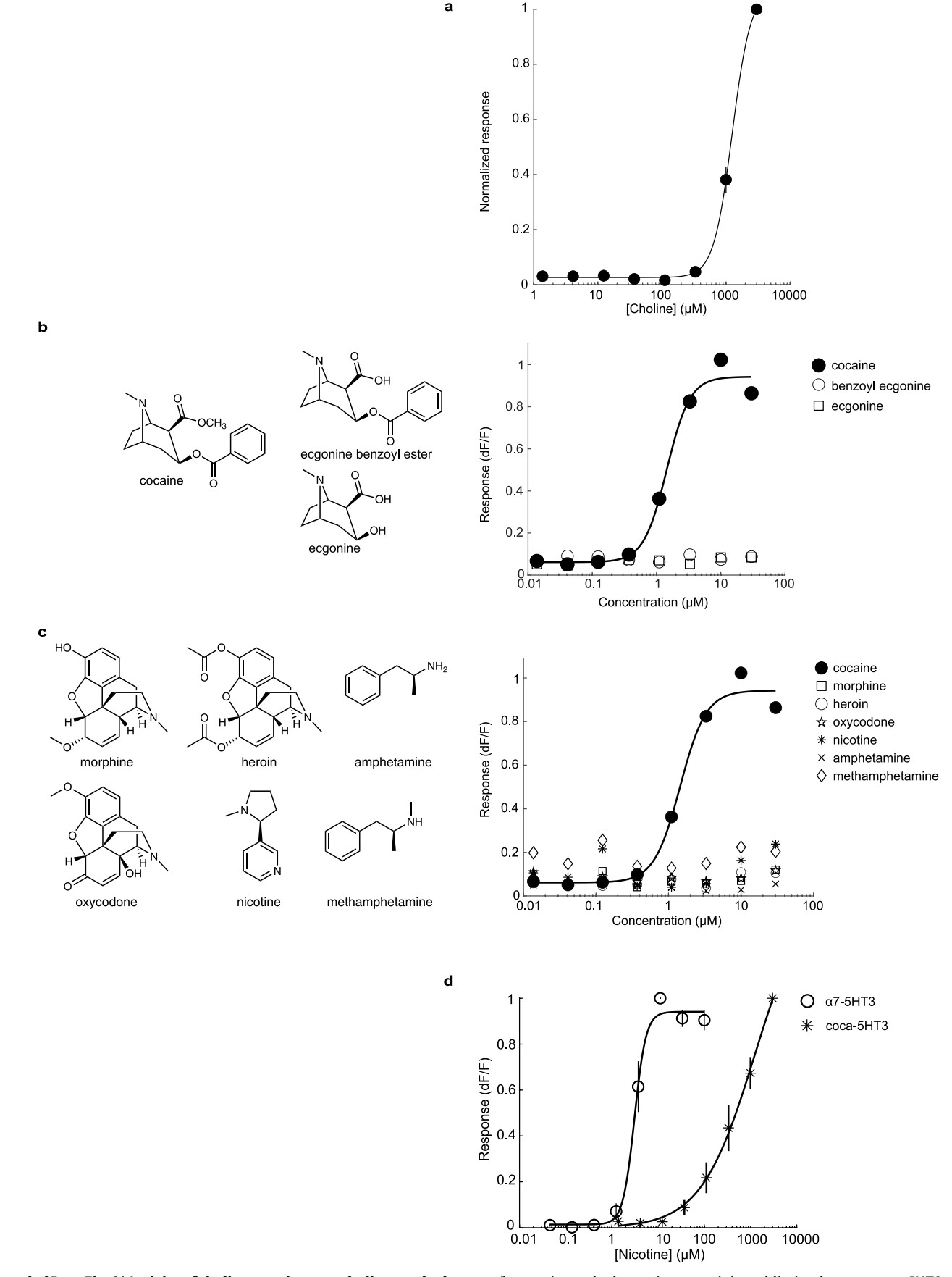

**Extended Data Fig. 2 | Activity of choline, cocaine, metabolites, and other drugs on coca-5HT3. a**, Dose response of choline on coca-5HT3 chimeric ion channel (n = 4 separate dose responses). **b**, Dose response for cocaine and two cocaine metabolites on coca-5HT3 (n = 1 dose response each). **c**, Dose response for cocaine and other amine-containing addictive drugs on coca-5HT3 (n = 1 dose response each). **d**, Comparison of dose responses for nicotine on unmodified α7-5HT3 (n = 6 separate dose responses) and coca-5HT3 (n = 3 separate dose responses). Data is shown as mean ± sem.

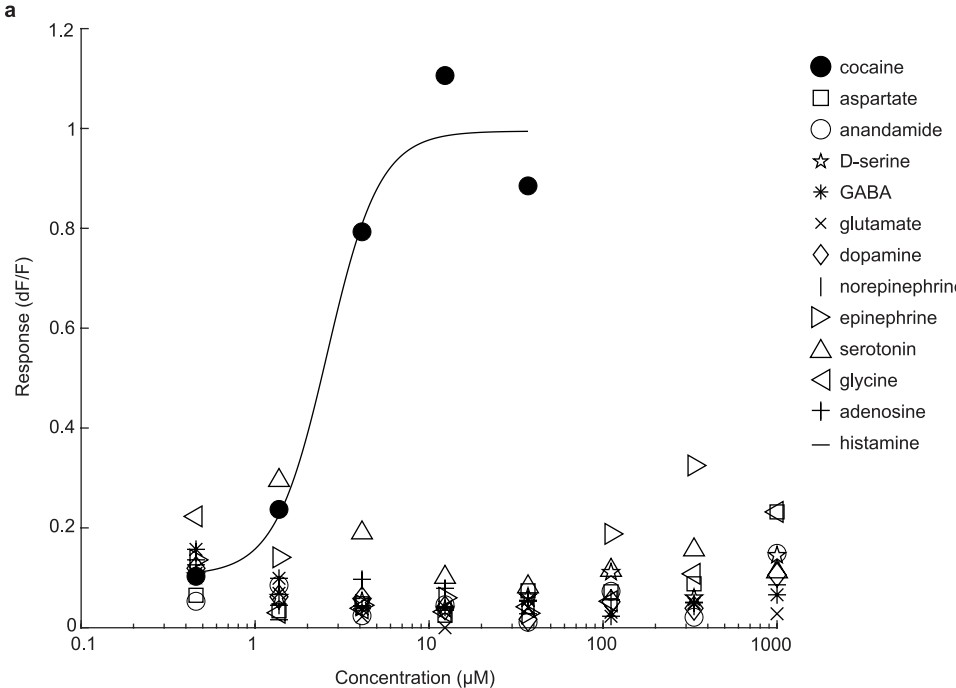

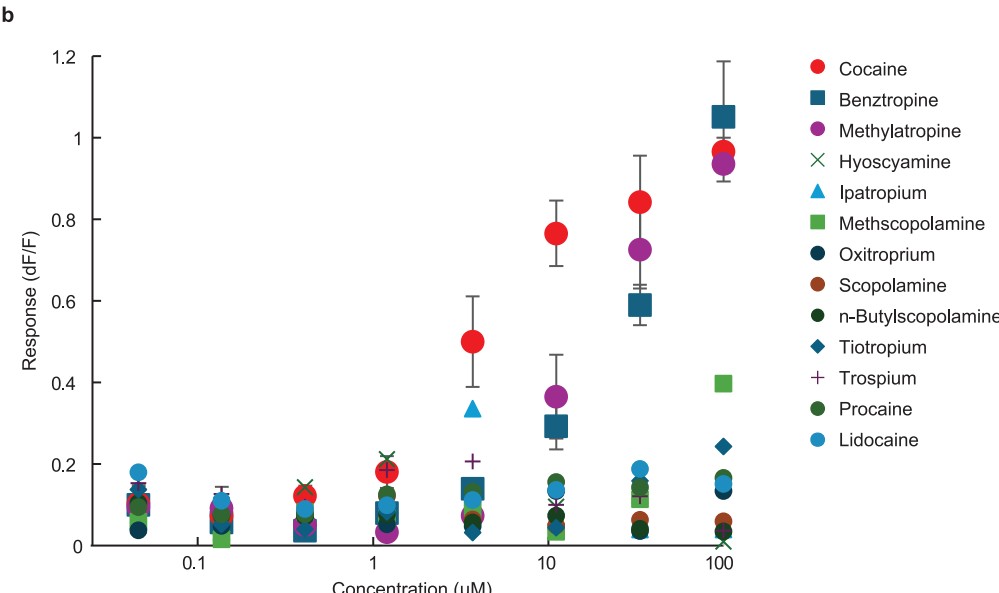

**Extended Data Fig. 3 | Activity of cocaine and physiologically relevant amines on coca-5HT3. a**, Endogenous amines (n = 1 dose response per amine). **b**, Clinically used drugs with structural similarity to cocaine and two common anaesthetics (cocaine, n = 5 separate dose responses; benztropine, n = 3 separate dose responses; methylatropine, n = 3 separate dose responses; all other drugs, n = 1 dose response). Data is shown as mean ± sem.

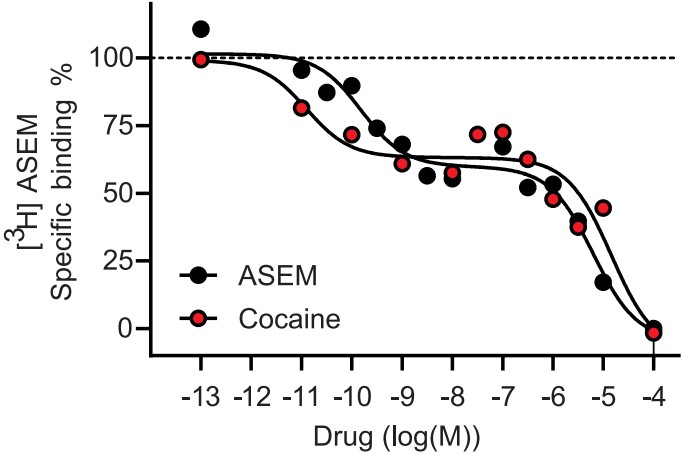

**Extended Data Fig. 4 | Competitive binding of cocaine and ASEM against [³H]ASEM binding in membranes from cells expressing coca-GlyR.** Dose-response curves were fitted using a two-site binding model. Data are shown as mean (n = 3 replicate curves for ASEM and n = 2 replicate curves for cocaine).

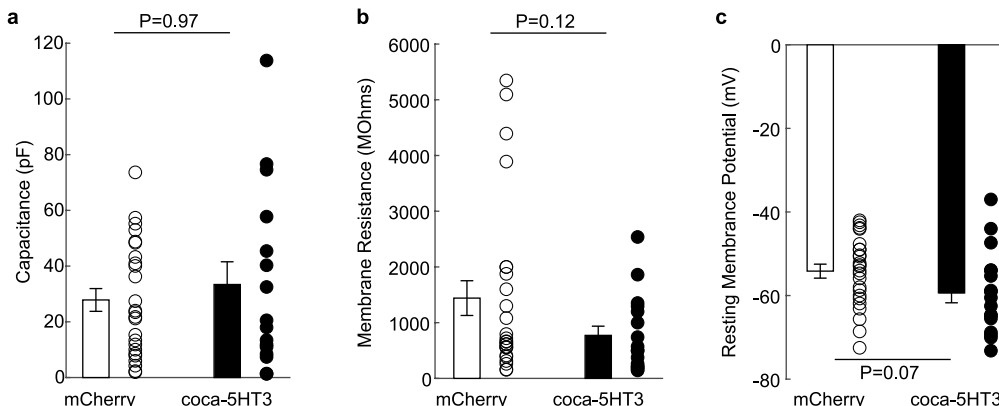

**Extended Data Fig. 5 | Membrane properties of LHb neurons expressing coca-5HT3.** LHb neurons from mice expressing coca-5HT3-IRES-mCherry (n = 16 cells) or negative control mCherry (n = 25 cells) showing **a**, Capacitance (Shapiro Wilk normality test P < 0.05, two-sided Mann-Whitney Rank Sum Test), **b**, Membrane Resistance (Shapiro Wilk normality test P < 0.05, two-sided Mann-Whitney Rank Sum Test), **c**, and Resting Membrane Potential (Shapiro Wilk normality test P = 0.97, two-sided t-test). Data is shown as mean ± sem, statistical tests are two-sided.

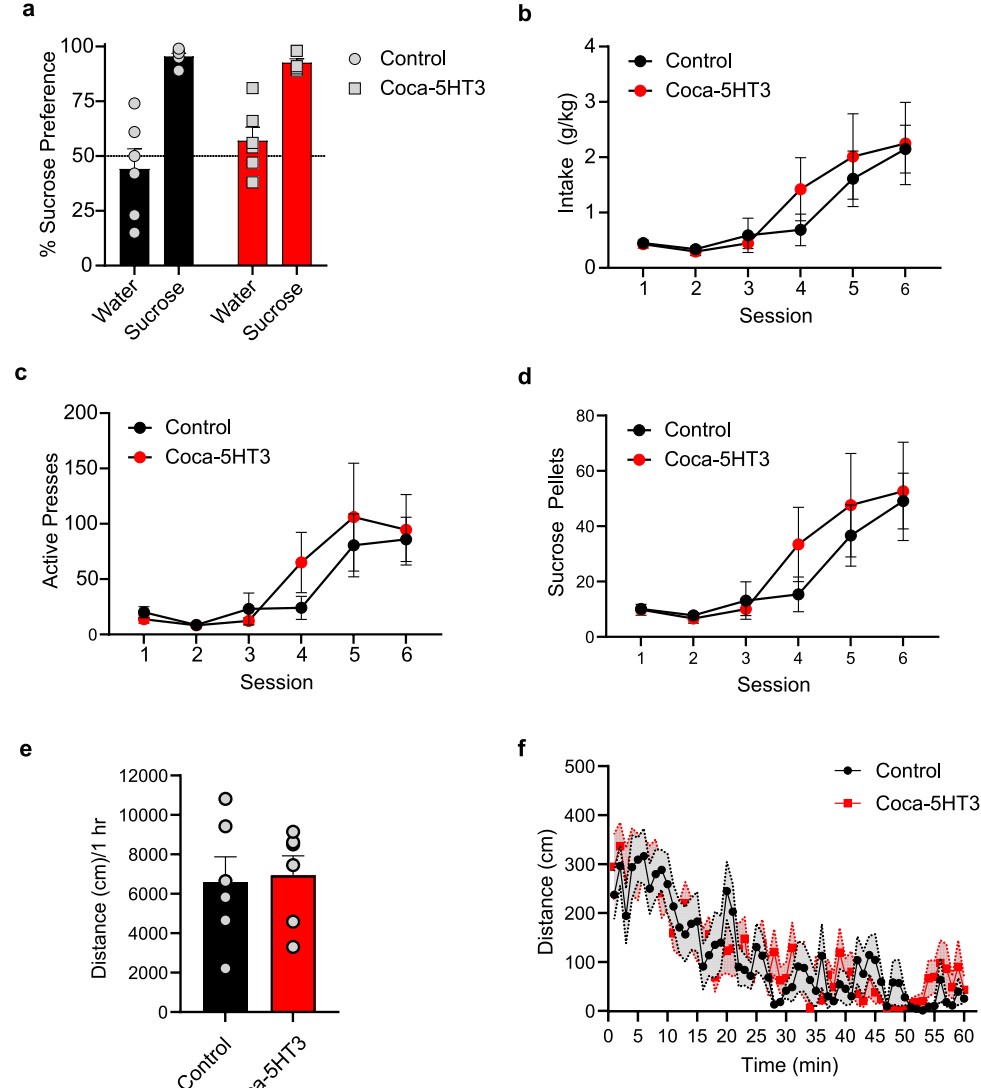

**Extended Data Fig. 6 | Control experiments to validate lack of motivation effects of coca-5HT3 LHb expression in the absence of cocaine. a**, Sucrose (1%) drinking preference in rats with coca-5HT3 expression in LHb (n = 6 rats) and in controls (n = 6 rats). **b**, Total sucrose intake (g/kg) (two-way RM ANOVA; transgene main effect (F(1, 11) = 0.18, p = 0.68), session main effect (F(5, 55) = 13.41, p < 0.001), transgene × session interaction (F(5, 55) = 0.61, p = 0.69)), (**c**) active lever presses (two-way RM ANOVA; transgene main effect (F(1, 11) = 0.24, p = 0.63), session main effect (F(5, 55) = 10.37, p < 0.001), transgene × session interaction (F(5, 55) = 0.72, p = 0.61)), and (**d**) number of sucrose pellets delivered during operant experiments in rats with coca-5HT3 expression in LHb (n = 5 rats) and in controls (n = 8 rats). All experiments are in the absence of cocaine. **e,f**, Open-field locomotor activity in rats with coca-5HT3 expression in LHb (n = 6 rats) and in controls (n = 6 rats) showing (**e**) total distance travelled over the 1 h session (two-sided unpaired t-test, t = 0.21; p = 0.84) and (**f**) within session time course of distance travelled. Experiment is in the absence of cocaine. Data is shown as mean ± sem.

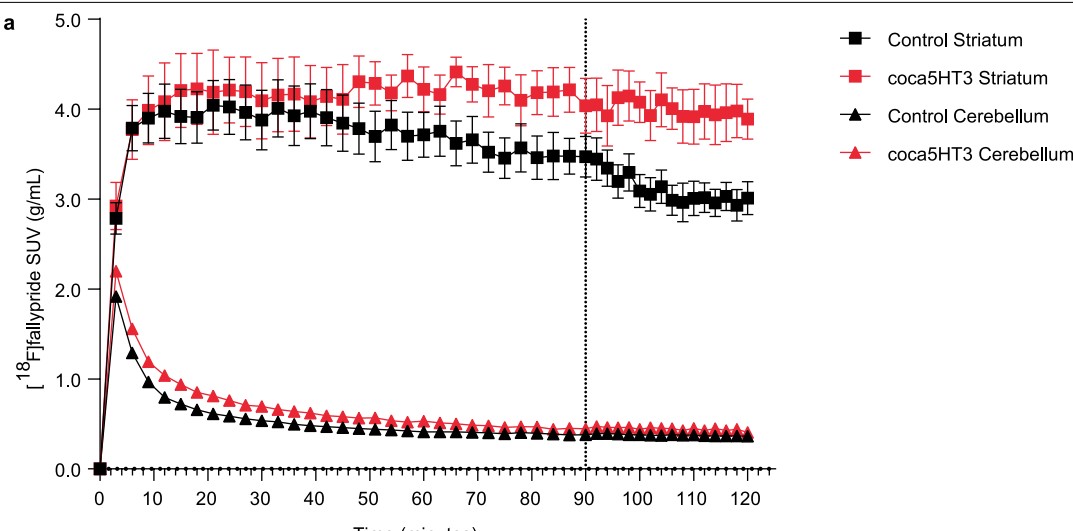

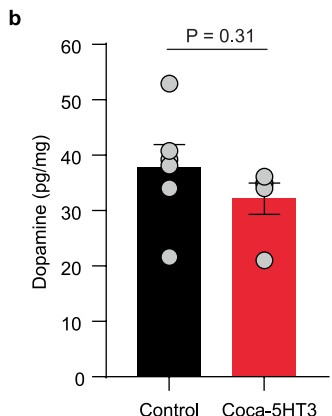

**Extended Data Fig. 7 | Dopamine levels for cocaine treatment and at baseline. a**, Time-activity curves of [$^{18}$F]fallypride binding (SUV; standardized uptake value) in the ventral striatum and cerebellum for control (n = 9 rats) and coca-5HT3 expressing rats (n = 9 rats). **b**, Baseline striatal dopamine concentrations in rats with coca-5HT3 expression in LHb (n = 6 rats) and in controls (n = 6 rats) in the absence of cocaine (unpaired t-test, P = 0.31). Data is shown as mean ± sem.

**a**

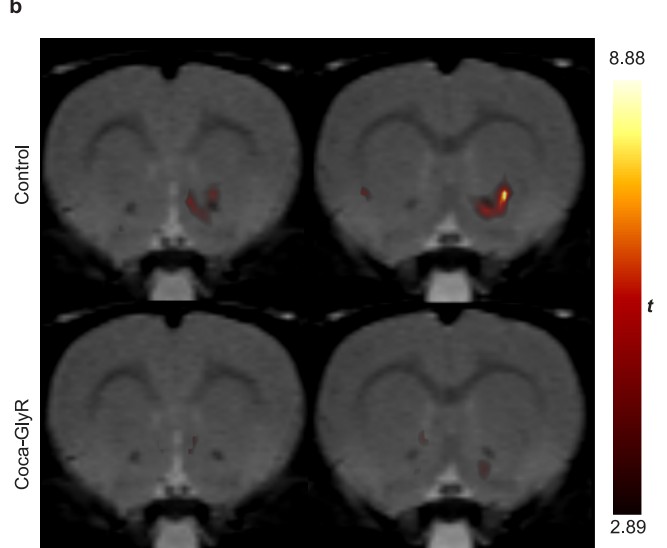

AAV1-Syn::Flex-coca-GlyR
Ventral tegmental area
TH-Cre rat

3 weeks

[¹⁸F]fallypride
+Cocaine 3 mg/kg
@ 90 minutes

PET scan

Analysis

**b**

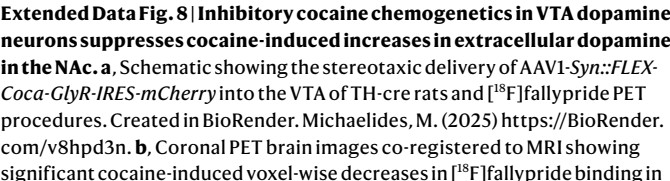

**c**

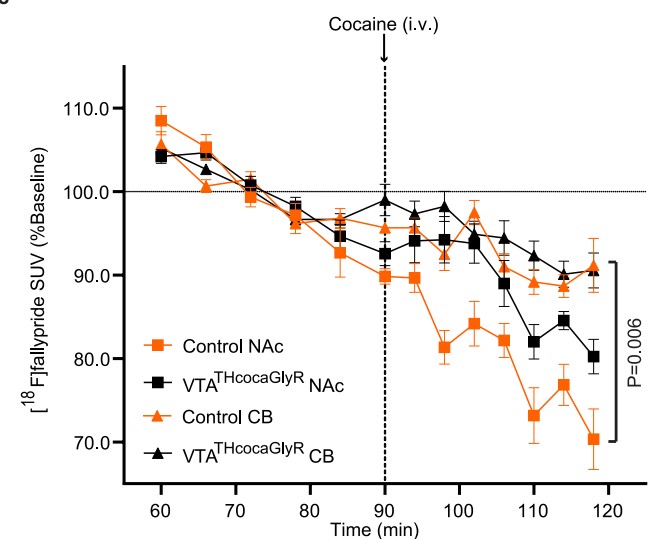

**Extended Data Fig. 8 | Inhibitory cocaine chemogenetics in VTA dopamine neurons suppresses cocaine-induced increases in extracellular dopamine in the NAc. a**, Schematic showing the stereotaxic delivery of AAV1-*Syn::FLEX-Coca-GlyR-IRES-mCherry* into the VTA of TH-cre rats and [¹⁸F]fallypride PET procedures. Created in BioRender. Michaelides, M. (2025) https://BioRender.com/v8hpd3n. **b**, Coronal PET brain images co-registered to MRI showing significant cocaine-induced voxel-wise decreases in [¹⁸F]fallypride binding in the ventral striatum of control rats (n = 6 rats) and in rats with coca-GlyR in VTA dopamine neurons (n = 9 rats). **c**, Time-activity curves of [¹⁸F]fallypride binding (SUV; standardized uptake value) in the ventral striatum (squares) and cerebellum (triangles) calculated as the percent of baseline (60–90 min) for control (n = 6 rats, black) and coca-GlyR rats (n = 9 rats, orange) showing that cocaine selectively displaces [¹⁸F]fallypride binding (i.e., increases dopamine) in the ventral striatum of control rats. Data is shown as mean ± sem.

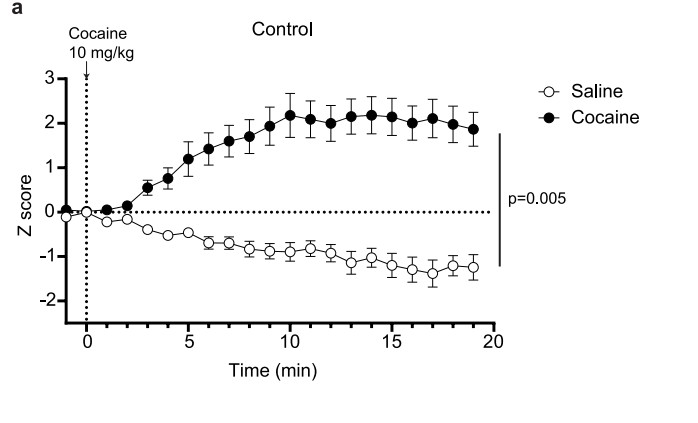

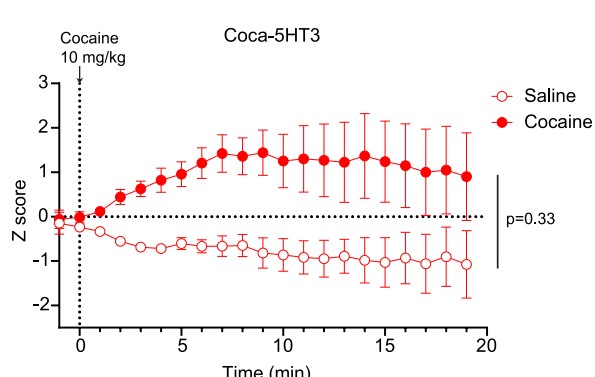

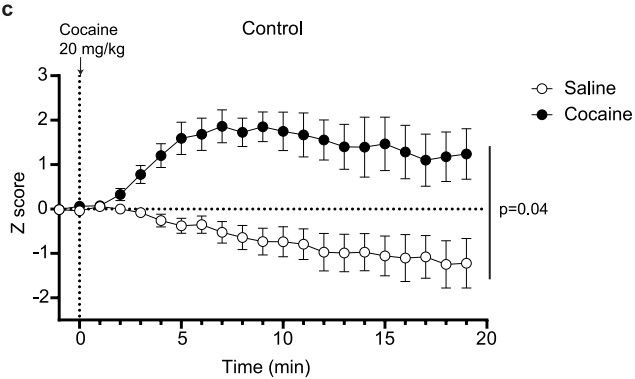

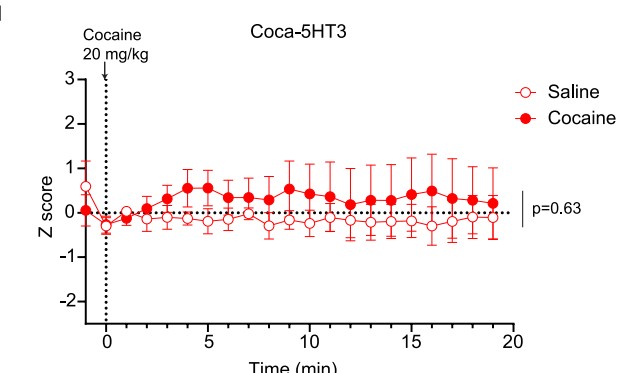

**Extended Data Fig. 9 | Influence of LHb coca-5HT3 expression on cocaine-induced dopamine levels in NAc. a-d**, Time course (z-score) of GRAB$_{DA}$ response in control rats treated with saline or cocaine (**a**, **c**) (10 mg/kg: two-way RM ANOVA; treatment main effect (F(1, 5) = 32.2, p = 0.002), time main effect (F(2.6, 13.1) = 4.56, p = 0.2), treatment × time interaction (F(1.7, 8.8) = 21.45, p = 0.005, 20 mg/kg: two-way RM ANOVA; treatment main effect (F(1, 5) = 13.61, p = 0.014), time main effect (F(1.4, 7.1) = 2.3, p = 0.17), treatment × time interaction (F(1.4, 7.2) = 5.09, p = 0.04) and coca-5HT3 rats treated with saline or cocaine (**b**, **d**) (10 mg/kg: two-way RM ANOVA; treatment main effect (F(1, 4) = 5.05, p = 0.08), time main effect (F(1.4, 5.6) = 1.12, p = 0.36), treatment × time interaction (F(1.1, 8.8) = 1.24, p = 0.33, 20 mg/kg: two-way RM ANOVA; treatment main effect (F(1, 4) = 0.41, p = 0.55), time main effect (F(2.4, 9.7) = 0.52, p = 0.63), treatment × time interaction (F(1.3, 5.4) = 0.37, p = 0.63). GRAB$_{DA}$ response in the medial nucleus accumbens shell measured using fibre photometry in rats with coca-5HT3 in the LHb (n = 5 rats) and in controls (n = 6 rats). Data is shown as mean ± sem.

**a**

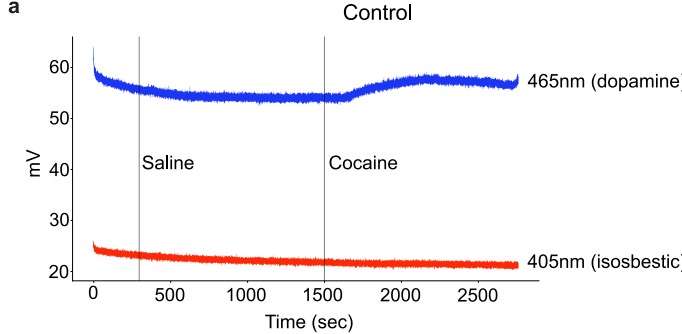

**b**

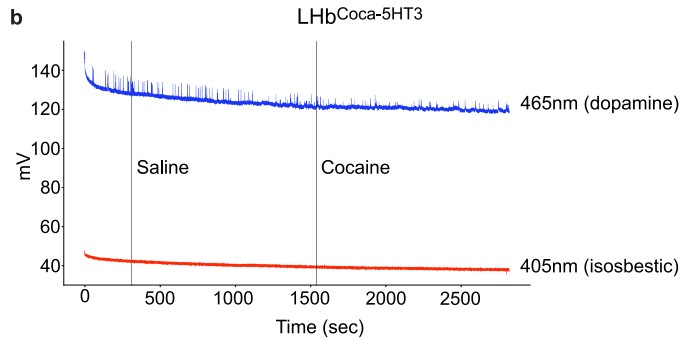

**Extended Data Fig. 10 | a,b,** Raw photometry data from a representative control (**a**) and LHb-coca-5HT3 rat (**b**) in response to saline and 5 mg/kg cocaine injections. The blue trace shows the 465 nm signal (dopamine), and the red trace shows the 405 nm isosbestic channel.

| | |
|---|---|

# Reporting Summary

## Statistics

For all statistical analyses, confirm that the following items are present in the figure legend, table legend, main text, or Methods section.

| n/a | Confirmed | |
|---|---|---|
| ☐ | ☒ | The exact sample size (*n*) for each experimental group/condition, given as a discrete number and unit of measurement |
| ☐ | ☒ | A statement on whether measurements were taken from distinct samples or whether the same sample was measured repeatedly |
| ☐ | ☒ | The statistical test(s) used AND whether they are one- or two-sided *Only common tests should be described solely by name; describe more complex techniques in the Methods section.* |
| ☐ | ☒ | A description of all covariates tested |
| ☐ | ☒ | A description of any assumptions or corrections, such as tests of normality and adjustment for multiple comparisons |
| ☐ | ☒ | A full description of the statistical parameters including central tendency (e.g. means) or other basic estimates (e.g. regression coefficient) AND variation (e.g. standard deviation) or associated estimates of uncertainty (e.g. confidence intervals) |
| ☐ | ☒ | For null hypothesis testing, the test statistic (e.g. *F*, *t*, *r*) with confidence intervals, effect sizes, degrees of freedom and *P* value noted *Give P values as exact values whenever suitable.* |
| ☒ | ☐ | For Bayesian analysis, information on the choice of priors and Markov chain Monte Carlo settings |
| ☒ | ☐ | For hierarchical and complex designs, identification of the appropriate level for tests and full reporting of outcomes |
| ☒ | ☐ | Estimates of effect sizes (e.g. Cohen's *d*, Pearson's *r*), indicating how they were calculated |

*Our web collection on statistics for biologists contains articles on many of the points above.*

## Software and code

Policy information about availability of computer code

| | |
|---|---|
| Data collection | MED-PC v.4.2 (Med Associates, Self-administration), Tucker Davis Technologies Synapse Software v. 95-44132P (Photometry), Nucline NanoScan 3.04.025 (Mediso, PET), LAS X 3.4 (Leica, Immunohistochemistry), pClamp 11. |
| Data analysis | Microsoft Excel 2024 and 2025, Graphpad Prism 10, MatlabR2016 & MatlabR2023b, Statistical Parametric Mapping (SPM12), PMOD v4.1, Sigma Plot 11.0 |

For manuscripts utilizing custom algorithms or software that are central to the research but not yet described in published literature, software must be made available to editors and reviewers. We strongly encourage code deposition in a community repository (e.g. GitHub). See the Nature Portfolio guidelines for submitting code & software for further information.

## Data

Policy information about availability of data

All manuscripts must include a data availability statement. This statement should provide the following information, where applicable:
- Accession codes, unique identifiers, or web links for publicly available datasets
- A description of any restrictions on data availability
- For clinical datasets or third party data, please ensure that the statement adheres to our policy

All data supporting the in vivo findings are available within this article and in the extended data. DNA constructs are available from Addgene.com under Materials Transfer Agreement. Data for model of cocaine bound to AChBP is available at PDB: 2pgz.

## Research involving human participants, their data, or biological material

Policy information about studies with [human participants or human data](). See also policy information about [sex, gender (identity/presentation), and sexual orientation]() and [race, ethnicity and racism]().

| | |
|---|---|
| Reporting on sex and gender | N/A |
| Reporting on race, ethnicity, or other socially relevant groupings | N/A |
| Population characteristics | N/A |
| Recruitment | N/A |
| Ethics oversight | N/A |

Note that full information on the approval of the study protocol must also be provided in the manuscript.

# Field-specific reporting

Please select the one below that is the best fit for your research. If you are not sure, read the appropriate sections before making your selection.

☒ Life sciences          ☐ Behavioural & social sciences          ☐ Ecological, evolutionary & environmental sciences

For a reference copy of the document with all sections, see [nature.com/documents/nr-reporting-summary-flat.pdf](http://nature.com/documents/nr-reporting-summary-flat.pdf)

# Life sciences study design

All studies must disclose on these points even when the disclosure is negative.

| | |
|---|---|
| Sample size | Sample sizes were estimated using G*Power or based on experience from past work. |
| Data exclusions | Rats were excluded if they lost catheter patency or due to lack of AAV transduction. Self-administration data were excluded for sessions where the infusion line disconnected from the rat's catheter port. |
| Replication | Detailed methods are provided to aid in replication by others. Behavioral, imaging and photometry experiments were replicated in at least 2 or more cohorts and by more than one experimenter. In vitro assays (binding, ELISA) were performed with at least two replicates and all replicate attempts were successful. |
| Randomization | Rats were randomly assigned to experimental groups and treatment conditions. Samples for in vitro assays (e.g. binding, ELISA) were also randomized with respect to treatment conditions. |
| Blinding | Experimenters were not blinded to group allocation during data collection for behavioral, imaging, and photometry experiments. For behavioral and photometry experiments data were collected by one researcher and analyzed by another. Data were analyzed blind if applicable but experimenters were always aware of the conditions. Immunohistochemistry experiments were performed blinded. |

# Reporting for specific materials, systems and methods

We require information from authors about some types of materials, experimental systems and methods used in many studies. Here, indicate whether each material, system or method listed is relevant to your study. If you are not sure if a list item applies to your research, read the appropriate section before selecting a response.

### Materials & experimental systems

| n/a | Involved in the study |
|---|---|
| ☐ | ☒ Antibodies |
| ☐ | ☒ Eukaryotic cell lines |
| ☒ | ☐ Palaeontology and archaeology |
| ☐ | ☒ Animals and other organisms |
| ☒ | ☐ Clinical data |
| ☒ | ☐ Dual use research of concern |
| ☒ | ☐ Plants |

### Methods

| n/a | Involved in the study |
|---|---|
| ☒ | ☐ ChIP-seq |
| ☒ | ☐ Flow cytometry |
| ☒ | ☐ MRI-based neuroimaging |

# Antibodies

| | |
|---|---|
| Antibodies used | rat monoclonal anti-mCherry (Clone 16D7, Invitrogen M11217), chicken polyclonal anti-GFP (Abcam ab13970), goat-anti rat immunoglobulin G (IgG) (H+L) crossabsorbed, Alexa Fluor 594 (Invitrogen, A11007), goat-anti chicken immunoglobulin Y (IgY) H&L (Alexa Fluor 488) (Abcam ab150173) |
| Validation | Primary antibodies were validated for immunohistochemistry by the specific vendor and confirmed in our lab where they showed selective immunolabeling in AAV targeted sites using AAVs with established fluorescent reporters. <br><br> rat monoclonal anti-mCherry (Clone 16D7, Invitrogen M11217): <br> https://www.thermofisher.com/antibody/product/mCherry-Antibody-clone-16D7-Monoclonal/M11217 <br><br> chicken polyclonal anti-GFP (Abcam ab13970): <br> https://www.abcam.com/en-us/products/primary-antibodies/gfp-antibody-ab13970 <br><br> goat-anti rat immunoglobulin G (IgG) (H+L) crossabsorbed, Alexa Fluor 594 (Invitrogen, A11007): <br> https://www.thermofisher.com/antibody/product/Goat-anti-Rat-IgG-H-L-Cross-Adsorbed-Secondary-Antibody-Polyclonal/A-11077 <br><br> goat-anti chicken immunoglobulin Y (IgY) H&L (Alexa Fluor 488) (Abcam ab150173): <br> https://www.abcam.com/en-us/products/secondary-antibodies/goat-chicken-igy-h-l-alexa-fluor-488-ab150169 |

# Eukaryotic cell lines

Policy information about cell lines and Sex and Gender in Research

| | |
|---|---|
| Cell line source(s) | ATCC CRL-1573 cells (HEK-293, passages 40-49) |
| Authentication | Cell lines were not authenticated after receipt from ATCC |
| Mycoplasma contamination | Cell lines were tested for mycoplasma contamination. If contaminated cultures were detected, then they were not used for experiments. |
| Commonly misidentified lines (See ICLAC register) | HEK cells can be misidentified as HeLa cells, but our lab received CRL-1573 HEK 293 cells directly from ATCC. We do not have HeLa cells in our lab. |

# Animals and other research organisms

Policy information about studies involving animals; ARRIVE guidelines recommended for reporting animal research, and Sex and Gender in Research

| | |
|---|---|
| Laboratory animals | Adult (8-10 weeks) Long Evans (CrL:Le) #006 rats, TH-cre rats,  adult (8-10 weeks) Sprague Dawley (Crl:SD) #400 rats, C57Bl/6J mice (male, 4-8 weeks) |
| Wild animals | The study did not involve wild animals |
| Reporting on sex | We used male rats for all experiments unless specified in the methods. |
| Field-collected samples | The study did not involve animals collected from the field |
| Ethics oversight | Experiments and procedures complied with ethical regulations for animal testing and research, followed the NIH guidelines and were approved by the NIDA animal care and use committee. Experiments and procedures complied with ethical regulations for animal testing and research and were approved by the Janelia Research Campus and UCSD animal care and use committees. |

Note that full information on the approval of the study protocol must also be provided in the manuscript.

# Plants

| | |
|---|---|
| Seed stocks | *Report on the source of all seed stocks or other plant material used. If applicable, state the seed stock centre and catalogue number. If plant specimens were collected from the field, describe the collection location, date and sampling procedures.* |
| Novel plant genotypes | *Describe the methods by which all novel plant genotypes were produced. This includes those generated by transgenic approaches, gene editing, chemical/radiation-based mutagenesis and hybridization. For transgenic lines, describe the transformation method, the number of independent lines analyzed and the generation upon which experiments were performed. For gene-edited lines, describe the editor used, the endogenous sequence targeted for editing, the targeting guide RNA sequence (if applicable) and how the editor was applied.* |
| Authentication | *Describe any authentication procedures for each seed stock used or novel genotype generated. Describe any experiments used to assess the effect of a mutation and, where applicable, how potential secondary effects (e.g. second site T-DNA insertions, mosiacism, off-target gene editing) were examined.* |

