## [Peer Review File · Nature]

Cocaine chemogenetics blunts drug-seeking by synthetic physiology

Corresponding Author: Professor Scott Sternson

Parts of this Peer Review File have been redacted as indicated to maintain confidentiality.

Version 0:

Reviewer comments:

Referee #1

(Remarks to the Author)

Negative feedback chemogenetics blunts drug-seeking by synthetic physiology

This is an interesting paper reporting the development of a new chemogenetic tool (coca-5HT3) that enables the direct stimulation of cells (via a modified cationic channel) by cocaine. It is postposed that coca-5HT3 can be used to stimulate cells in the brain that are otherwise inhibited by cocaine, thereby inducing a “negative feedback” cycle. It is suggested negative feedback cycle can be used to investigate the contributions of the targeted populations of cells in regulating behavioral responses to cocaine.

1. The concept of “negative feedback” in the context of cocaine actions on the habenula is elegant and interesting. However, it is also a marked oversimplification of the actual effects of cocaine on neuronal activity in this brain region (and on many other regions). Cocaine elicits transient decreases in habenula activity that correlate with increased cocaine-seeking behaviors, as described by the authors. However, this effect is very quickly followed (order of minutes) by persistent increases in habenula activity. Thus, by expressing an excitatory ion channel in the habenula that is a directly stimulated by cocaine, the initial inhibitory effect of cocaine on habenula can be blocked (although the precise population of habenula neurons usually impacted by cocaine may not be targeted). Thus, the negative feedback loop noted by the authors in the title of the paper is established. However, this same manipulation also recapitulates the delayed excitatory effect of cocaine in the same habenula cells. In this case, the system actually establishes a “positive feedback” loop.

2. In the opening paragraph, it is stated that optogenetics and chemogenetics “use non-physiological open-loop neural perturbations”. It is not clear to this reviewer that this statement is entirely accurate. Optogenetic manipulations can recapitulate natural firing patterns of cells. Chemogenetics (at least the DREADD variety) “primes” cells for activity by modulating intracellular second messenger systems, and hence modulates endogenous firing patterns. Optogenetic and chemogenetics (particularly optogenetics) can be used to reverse drug-induced changes in neural activity by imposing natural/physiological firing patterns on cells. Moreover, “closed loop” systems combining cell imaging with optogenetics have been developed, which can impose patterns of neural activity in near real-time. Finally, it is unclear “why” optogenetics and chemogenetics need to depend on [addictive] drug exposures (pharmacokinetics of the drug). Instead, optogenetics and chemogenetics should depend on changes in cellular activity induced by drug exposure levels (pharmacodynamics of the drug), which they do. Overall, the drawbacks of optogenetics and chemogenetics highlighted in the opening paragraph appear somewhat forced, and the paragraph should be re-structured to instead better describe the unique benefits of coca-5HT3 and similar systems.

3. Although not statistically significant in the overall ANOVA, there was a clear trend for rates of food responding to be lower in the coca-5HT3 rats relative to control rats, even though they are cocaine-naïve. This is particularly apparent during the initial 1-5 days of training (under the FR1 schedule) but the trend persisted throughout testing. This could suggest that expression of coca-5HT3 in the habenula impacts operant responding independent of cocaine treatment. Thus, a broader range of behavioral assessments should be performed to confirm that coca-5HT3 expression in the habenula does not impact behavior under baseline conditions.

4. Related to the above, baseline electrophysiological properties of coca-5HT3-expressing habenula neurons should be

assessed (resting membrane potential; spontaneous action potential frequency; rheobase, etc.).

5. Likewise, the effects of coca-5HT3 on cell responses in the RMTg and/or other brain regions that receive synaptic input from the LHb should be assessed.
6. Are there any changes in the self-administration of drugs of abuse other than cocaine (nicotine or heroin, for example) in coca-5HT3-expressing animals? Likewise, does coca-5HT3 expression alter responding for a high-value natural reinforcer like sucrose? This is important to confirm drug/behavioral selectivity.
7. Habenula inactivation is known to induce hyperactivity. Cocaine also induces hyperactivity. Are coca-5HT3 animals so hyperactive that they are unable to respond for cocaine?
8. Other addiction-related behaviors known to be modulated to cocaine and that may involve habenula neurons should be assessed. For example, is the stimulant effect of cocaine altered, or cocaine-induced locomotor sensitization?
9. The PET imaging and GRABDA photometry are interesting. However, but are “relative” measures of dopamine signaling. It is important to confirm that coca-5HT3 expression in the habenula does not impact “absolute” dopamine levels in the accumbens.
10. What are the effects of coca-GlyR on cocaine-related behaviors and dopamine signaling in the accumbens?

Referee #2

(Remarks to the Author)

This is an elegant study that describes the approaches used to generate and utilize a new tool that will be highly beneficial to investigators in the drug dependence field. The authors provide an excellent series of experiments that shows the characterization of these chimeric cocaine-selective receptors and then demonstrate their utility in in vivo models. Altogether, this is exciting work. That said, there are some issues that need to be addressed. Much of these issues are likely due to writing within the word limit of the journal. Therefore many of these issues are associated with clarity of the specific methods.

Major Comments:

1. Page 4: The authors reference the pharmacophore of shared similarity between nAChR agonists and addictive drugs but Extended data 1b looks to have not made it into the submission (only the chimeric schematic of extended data fig. 1a is present, and it does not include 1b).
2. Page 6, last paragraph: The authors used viral methods to express Coca-5HT3 for electrophysiology assays. However, they do not discuss if these are cultured neurons or in slices (the methods do specify hippocampal cultures but this should be clear in reading the Results section).
3. Why were rheobase measurements not completed for coca-5HT3 as they were with coca-GlyR?
4. Figure 2d shows a concentration-dependent increase in neuronal firing. Why is there no mean data associated with this observation? Were these current clamp recordings not replicated in a manner that would facilitate an analysis of the mean? In 2c, the authors show mean data for cocaine-induced depolarization of hippocampal neurons but don't provide representative traces. This also occurs with Figure 2e. In both cases, representative waveforms should also accompany the mean data. If there is not sufficient space, then this could be put in the extended data.
5. The only photometry data present is the processed Z-scores of control and coca-5HT3-injected rats. The authors should provide sample data of raw signal and isosbestic signal traces. Many investigators inject GRABDA or dLight and then implant optical cannulas. Given that the authors implanted cannulas that were coated in GRABDA virus, it would be very important to show what the raw photometry signals look like. Given that the authors used workflows from the Lerner lab (GuPPy perhaps?), some of these data will have plots generated already and these can be added to a revised figure.
6. The results section associated with the photometry is also vague. How was the cocaine treatment conducted? Was this done in a simple open field, rats were allowed to habituate, and then the rats were sc/ip injected with cocaine and placed back in the arena during the recording session? Once again, much of this is explained in the methods but some of these details should be provided in the results section. This also highlights the need for providing traces as there were likely handling artifacts for injecting the cocaine during the photometry sessions.
7. Specific for the photometry measurements, the authors should provide the frequency at which the recordings were set. The laser power was provided but it was not clear if this was for both the GRABDA signal and for the isosbestic.
8. No electrophysiological methods were provided for the recordings of the hippocampal neurons. I assume the same amplifier/digitizer setup was used as with the HEK293 electrophysiology. However, with cultured neurons some investigators use specific extracellular buffers that differ from what is used for HEK cells. This should be provided. Also, there needs to be a discussion of what protocol was used to measure Rheobase. Finally, the authors should describe what methods they used to calculate specific cell parameters (capacitance, resistance, input resistance, and resting potential).
9. The in vivo utilization of coca-5HT3 provides functional validation for these chimeric proteins in a behavioral model. However, the authors developed and characterized the coca-GlyR protein as well. As the functional validation shows, cocaine will activate the coca-5HT3 receptors robustly but will inactivate the coca-GlyR. Given the behavioral result of 'selective' activation of coca-5HT3 in the LHb, it would be extremely interesting to see if using coca-GlyR to impose the opposite effect would produce the reverse. Understandably, this would take significant time to recapitulate the cocaine IVSA with coca-GlyR but something that may be less time intensive would be to see if coca-GlyR would enhance NAc dopamine given that coca-5HT3 depressed NAc dopamine release.

Minor Comments:

1. Figure 3C: The number of days (or range in days) for food training, cocaine training, and dose response should be indicated in the schematic
2. Figure 4D: The authors should add a legend to describe the different data points.
3. Page 4: “ $\alpha 7$ nicotinic acetylcholine receptor (nAChR) ligand gated....” Is not the typically language used within the nAChR field. I know this was just a stepping stone to generate the protein(s) at the focus of this work. However, it would be more correct to rework this sentence to something like: “ $\alpha 7$ nicotinic acetylcholine receptors (nAChRs) are ligand gated ion channels that have been engineered....”
4. Page 5: references for ACh/nAChR cation- π binding should include additional references since this has been replicated in multiple labs and model systems. The original work that led to our current knowledge dates back to 1998 (Zhong et al., 1998, PNAS).
5. The authors show competition binding of cocaine for the coca-5HT3 receptor. Was the same assay completed for any of the $\alpha 7$ GlyR chimeric mutants?
6. In the legend of Figure 2(d) the author state that current injections were used to monitor membrane properties during the collection of current-clamp data. This needs to be detailed in the methods or extended data.

Referee #3

(Remarks to the Author)

Gomez et al developed novel engineered ionotropic receptors that respond selectively to cocaine, and use them to demonstrate what they refer to as “closed-loop” control of cocaine self-administration via manipulation of lateral habenula circuits influencing nucleus accumbens dopamine signaling. The paper offers two main contributions to the field.

The first is their strategy for engineering a drug (cocaine)-sensitive excitatory or inhibitory ionotropic receptor, and validating its function in vitro and in vivo to control certain self-administration behaviors, and downstream dopamine signaling.

The second and more important is the concept of ‘closed-loop’ control of neural circuits underlying cocaine intake. The idea of closed-loop control of maladaptive neural circuit activity is not novel per se, but to my knowledge this is the first test of the idea in the context of intake of a specific drug, here cocaine, and thus it represents a conceptual advance. This new idea could be experimentally employed to test circuit functions and control of behaviors in animals as done here, and it could potentially in future be implemented to control maladaptive behaviors in humans addicted to specific drugs like cocaine. They note this closed-loop approach could also be developed to intervene in other biological feedback systems such as those involving hormones or nutrients, which is an exciting potential future application.

However, the paper has some significant limitations as well.

The premise of closed-loop control of cocaine intake relies here upon a neural pathway (LHb midbrain GABA populations dopamine neurons) that is assumed, but not actually directly tested. It was surprising that given the focus on dopamine neurotransmission as an endpoint, the cocaine-sensitive receptor was not expressed in dopamine neurons themselves—this would be more direct test of the proposed mechanism of “closed loop control”.

Other issues exist regarding specificity of the presented data. They show that the receptor does not bind other addictive drugs or endogenous neurotransmitters, but drugs structurally similar to cocaine, some of which are used clinically like lidocaine or Novocain, were not tested. Anatomical specificity of presented behavioral and dopamine data should also be clarified. Does, as one would expect under the proposed framework, intra-LHb cocaine do the same things as systemic cocaine? Does cocaine (systemic or intracranially applied to the receptors) alter other behaviors controlled by LHb and its projections? Does PET show cocaine at relevant doses demonstrably binds the receptor in vivo?

Another feature of the paper that I have trouble with is the pattern of effects on cocaine self-administration behavior. Control and coca-5HT3 mice acquire self-administration in a “statistically equivalent manner,” which is hard to understand given the proposed “closed loop” mechanism—wouldn’t cocaine in these animals be expected to attain an aversive quality that would suppress self-administration? Lines 177-80, and 261-63 address this point and note that “coca-5HT3 does not significantly interfere with cocaine self-administration with a high infusion dose that does not require high instrumental responding to maintain exposure to reinforcing levels of cocaine”. However I do not understand this logic since effort required to obtain cocaine should not be especially relevant to the rewarding vs aversive properties of cocaine itself—the variables which are proposed to interface with cocaine intake in a closed-loop manner. Likewise, since only high effort cocaine intake is affected by the manipulation, and high dose cocaine intake is not affected, this seems to undercut the translational relevance of the approach, since an addicted human could override the intervention by taking a larger dose of cocaine.

More explicit information about the sex of experimental subjects in each experiment is also required—some appear to be only in males despite the information provided in the additional information provided for review saying all studies are in males and females.

Finally, the initial sections of the paper would benefit from concrete definition of conceptual terms such as “non-physiological open-loop neural perturbations”.

Version 1:

Reviewer comments:

Referee #1

(Remarks to the Author)

The authors have nicely addressed my concerns about the original version of the manuscript. They have generated new data that further validate their novel chemogenetic tool and help establish its specificity (i.e., activity only in the presence of cocaine). The new electrophysiological analyses are particularly helpful in this regard. I have no further comments for their consideration.

Referee #2

(Remarks to the Author)

The authors have done an excellent job in revising their manuscript. My only additional comments are minor.

Minor Comments:

1. Consistency of error bars (lines in panels 1f and 2h but T-bars in all other figures/panels).
2. The statistical test used in Figure 3 is not stated in the Figure 3 legend.
3. The statistical test used in Figure 4 is not stated in the Figure 4 legend.

Referee #3

(Remarks to the Author)

Gomez et al have responded to most of my primary concerns, including adding some new data and analyses that improve the paper. I still believe one major concern of Revs. 1 & 3 regarding interpretations of behavioral results remains, however. Additional minor adjustments are also required.

The major remaining concern regards interpretation of the cocaine self-administration findings as specifically involving a closed loop negative feedback mechanism. This is a major conceptual selling point of the paper, and it was previously critiqued by Rev.1 in their points 7&8. The authors in response now clarify this point somewhat, but I still believe this interpretation is on shaky ground with the current data alone supporting it. Rev.1 suggests other behavioral tests that could be conducted to bolster this claim, and the authors rebut their usefulness, but I tend to agree they may be necessary to resolve this important loose end in a main conceptual takeaway of the paper.

Specifically, I am concerned that their explanation of why LHb manipulations selectively alter cocaine intake at certain doses, but not at 0.5mg/kg during initial FR1/5 training, is weak, as it stands. They claim this reflects differences in (dopamine-mediated) motivation required to self-administer the drug at these doses. If so, a very useful comparison condition would be to test a non-cocaine self-administration related behavior (at least locomotion), conducted in the presence of experimenter-delivered cocaine. Food self-administration tested under different response requirements or reward magnitudes, they paralleling cocaine data on which effects were observed, would be best for testing their motivation-related interpretation. Adding such a control experiment would thus help disambiguate the impacts of motivation, pharmacology, reinforcement, negative feedback, and dopamine that are currently intermingled, and potentially undercutting of the premise of a closed loop negative feedback mechanism specific to cocaine taking.

Minor Points:

Line 261: "Each rat was tested 3 times (starting at 20, then 10, then 5 mg/kg) in randomized order". Clarify if doses were tested in fixed or randomized order.

-Line 552-3: "The tissue was then chunked removing the olfactory bulbs and coronally sectioning at the optic chiasm." For ELISA dopamine analysis: clarify what this means, and which tissue exactly was analyzed for DA levels

-Line 590: "Rats could commit a maximum of 60 infusions when tested at the 0.5 and 1 mg/kg unit doses but no infusion limit was present at other unit doses" Authors should note if any rats reached the 60 infusion limit, and any potential implications of this on the data overall.

-Please clarify for all experiments what strain and sex of animal was used, and acknowledge in the manuscript any potential implications of the multiple rat strains used, and exclusive use of males, on implications and generalizability of these findings.

-Please clarify in reporting checklist that females were not used in any experiments (or where they were used if they were).

-Clarify in main text that vectors targeting dopamine neurons in TH:Cre rats were cre-dependent

RESPONSE TO REVIEWERS

We thank the reviewers for their thoughtful comments about our manuscript. We address individual points below in red type, reviewer questions are in blue, and text quoted from the manuscript is in black.

Referee #1 (Remarks to the Author):

Negative feedback chemogenetics blunts drug-seeking by synthetic physiology

This is an interesting paper reporting the development of a new chemogenetic tool (coca-5HT3) that enables the direct stimulation of cells (via a modified cationic channel) by cocaine. It is postposed that coca-5HT3 can be used to stimulate cells in the brain that are otherwise inhibited by cocaine, thereby inducing a “negative feedback” cycle. It is suggested negative feedback cycle can be used to investigate the contributions of the targeted populations of cells in regulating behavioral responses to cocaine.

Based on the feedback from the editor: “...address the comments from Reviewer #1 on demonstrating that this does indeed represent a negative feedback loop.”, we want to describe here how we approach this before we respond to individual critiques and questions. In this study, we engineered an artificial negative feedback relationship between self-administered cocaine and cocaine self-administration (normally these have a positive feedback relationship). The characteristics of the negative feedback framework are that the chemogenetic perturbation needs to: 1) oppose the normal effect of the addictive drug and 2) be engaged only when the addictive drug is delivered (i.e., not affect the circuit in the absence of cocaine). To achieve this goal, we developed the first cocaine-gated ion channels as a tool to oppose cocaine-induced neural activity changes. Because the engineered ion channels are intrinsically sensitive to cocaine, this artificial opposing negative feedback should only show evidence of engaging the functional consequences of LHb activity in the presence of cocaine. We showed that the cocaine-induced chemogenetic perturbation negatively regulates self-administration of cocaine. We also show that the well-established cocaine-induced rise in dopamine is suppressed, as expected from our designed negative feedback mechanism. We interpreted Reviewer 1’s Questions 1-9 as focused on additional validation of the negative feedback chemogenetic framework by requesting more evidence and discussion of these two components. Because the functional effects on dopamine and cocaine self-administration clearly showed the capacity of this system to negatively regulate the positive cycle of cocaine ingestion and self-administration, the key remaining issue was further demonstrating that this was a feedback effect, i.e., solely due to the influence of cocaine vs an effect of the cocaine-gated ion channel in the absence of cocaine. Our new experiments show a lack of effects on food self-administration, basal dopamine, and basal locomotion and provide more evidence that the cocaine-gated ion channel does not affect cocaine-independent behaviors. This is in striking contrast to the strong effects to oppose cocaine self-administration, which is exactly what would be expected from the negative feedback mechanism around which we designed these new cocaine-gated ion channels.

1. The concept of “negative feedback” in the context of cocaine actions on the habenula is elegant and interesting. However, it is also a marked oversimplification of the actual effects of cocaine on neuronal activity in this brain region (and on many other regions). Cocaine elicits transient decreases in habenula

activity that correlate with increased cocaine-seeking behaviors, as described by the authors. However, this effect is very quickly followed (order of minutes) by persistent increases in habenula activity. Thus, by expressing an excitatory ion channel in the habenula that is a directly stimulated by cocaine, the initial inhibitory effect of cocaine on habenula can be blocked (although the precise population of habenula neurons usually impacted by cocaine may not be targeted). Thus, the negative feedback loop noted by the authors in the title of the paper is established. However, this same manipulation also recapitulates the delayed excitatory effect of cocaine in the same habenula cells. In this case, the system actually establishes a “positive feedback” loop.

We agree with the reviewer’s premise that the effects of cocaine on the LHB (and elsewhere) have complex dynamics. This was the primary inspiration for developing the negative feedback chemogenetic approach. The complex neural dynamics during addictive drug taking is associated with the rapid rise and clearance of cocaine in the brain. For example, Pan, et al 1991 (top left panel) show in rats that an intravenous dose of cocaine causes brain cocaine to rise over 5 min and fall substantially by 15 min (importantly [cocaine] drops below the EC50 of coca-5HT3). The neural dynamics reported for

FIG. 3. Cocaine concentrations in the dialysate from brain (A) and blood (B) following 7.5 mg/kg i.v. administration, n = 5. Active length of probe in the blood is 5 mm and in N Acc 2 mm; perfusion rate is 1.2 µl/min. The vertical bars represent the SEM. The maximum concentration of cocaine in the blood occurs at T = 0, since the drug is introduced directly into the bloodstream as a bolus injection. This cannot be measured by microdialysis but can be estimated by nonlinear regression as in Fig. 4.

LHB inhibition during intravenous cocaine administration in Jhou 2013 JNeurosci (2nd right panels A&B, G) follow a timecourse over 0-15 min that mirrors the rise and fall in brain cocaine concentration. The mechanism of action for these cocaine-gated chemogenetic receptors is intrinsically tied to the timecourse of cocaine exposure. During the cocaine exposure period, the excitation of coca-5HT3 corresponds to the negative feedback effect that the reviewer acknowledges. Note the peak cocaine concentration from Pan, et al is approximately 1.5-fold higher than the EC50 for coca-5HT3, so this channel is sensitive within the dynamic range of intravenous cocaine exposure. The delayed excitatory effect reported by Jhou ,et al., which occurs in less than half of the LHB cocaine-inhibited neurons, corresponds to a time window associated with cocaine clearance, and our chemogenetic receptors are not active when cocaine concentration falls (see the very rapid washout in seconds in the voltage clamp recordings in Fig. 2a). We also note that the later activity of LHB neurons reported by Jhou et al corresponds to a timeframe that has been associated with anti-rewarding effects of cocaine after its brain clearance (citations: 6,34,36 in the revised manuscript). Our approach is to induce this anti-reward effect opposing the cocaine-induced LHB inhibition during the cocaine-exposure period in a manner that is intrinsically linked to the rise and fall in cocaine concentration. This is the unique strength of our chemogenetic approach based on cocaine-gated ion channels as compared to traditional chemogenetics. To better clarify these points we have added the following two sentences on p. 7-8: “The LHB is normally inhibited by cocaine *in vivo*, in part *via* elevated dopamine acting on the dopamine type 2 (D2) receptor in LHB neurons³⁴, and LHB inhibition promotes cocaine self-administration³⁵. Cocaine-mediated LHB

inhibition is transient (0-15 min)³⁴, tracking the similarly short cocaine brain exposure timecourse⁶, and is associated with reward in this timeframe^{6,34,36}.” **On p. 13-14 (Discussion) we have also noted that:** “Following the rewarding spike in cocaine after intravenous administration, brain cocaine levels quickly fall within 15 min⁶, and after this, some LHb neurons show activation³⁴ and cocaine also becomes aversive³⁶. Cocaine-activation of coca-5HT3 in the LHb induces the anti-rewarding effect of LHb activation during the cocaine-exposure period, thus opposing high effort cocaine self-administration. This negative feedback perturbation is intrinsically linked to the rise and fall of the cocaine concentration. This is a consequence of the rapid activation of coca-5HT3 by cocaine and the rapid deactivation of coca-5HT3 upon cocaine removal.”

2. In the opening paragraph, it is stated that optogenetics and chemogenetics “use non-physiological open-loop neural perturbations”. It is not clear to this reviewer that this statement is entirely accurate. Optogenetic manipulations can recapitulate natural firing patterns of cells. Chemogenetics (at least the DREADD variety) “primes” cells for activity by modulating intracellular second messenger systems, and hence modulates endogenous firing patterns. Optogenetic and chemogenetics (particularly optogenetics) can be used to reverse drug-induced changes in neural activity by imposing natural/physiological firing patterns on cells. Moreover, “closed loop” systems combining cell imaging with optogenetics have been developed, which can impose patterns of neural activity in near real-time. Finally, it is unclear “why” optogenetics and chemogenetics need to depend on [addictive] drug exposures (pharmacokinetics of the drug). Instead, optogenetics and chemogenetics should depend on changes in cellular activity induced by drug exposure levels (pharmacodynamics of the drug), which they do. Overall, the drawbacks of optogenetics and chemogenetics highlighted in the opening paragraph appear somewhat forced, and the paragraph should be re-structured to instead better describe the unique benefits of coca-5HT3 and similar systems.

Our goal here was to highlight the unique characteristic of our approach, which is to directly yoke the chemogenetic perturbation precisely to addictive drug exposure. Other chemogenetic methods and optogenetic methods can indirectly alter the sensitivity of neurons during drug exposure, but this sensitivity follows the timecourse of a chemogenetic ligand such as clozapine. As the reviewer notes, in the case of traditional chemogenetics, the neurons may be more or less sensitive to neural inputs during addictive drug exposure, but this chemogenetic perturbation is not intrinsically tied to the drug exposure timecourse and the chemogenetic agonist almost always has a different timecourse than the variable addictive drug self-administration timecourse. For closed-loop optogenetics, neuron perturbations can be experimentally locked to a behavioral epoch or a measurement of a readout of some signal associated with neuron activity or neuromodulation, but there is still only an indirect association with the actual exposure level of the addictive drug. It is a general challenge with chemical feedback to know the precise timecourse of the chemical exposure at the cells of interest in vivo. The strength and conceptual advance of our new approach is that this timecourse is intrinsic to our chemogenetic strategy for neuromodulation. To better describe the benefits of coca-5HT3, we have modified our statements by highlighting the drug-yoked aspect of our approach relative to the more indirect control that is possible with traditional chemogenetics and closed loop optogenetics. We now write: “Traditional approaches for addiction research and therapy use open-loop neural perturbations, such as pharmacology and chemogenetics⁶, that are not directly coupled to the addictive drug brain concentrations, and even fast-timescale optogenetic or electrical stimulation are not yoked to changes in addictive drug exposure levels^{7,8}.” **We also added a new sentence to clarify the point: “Because the neural signaling pathways from addictive drugs are**

sensitive to the drug exposure timecourse, it is desirable to tailor neural perturbations to the temporal dynamics of addictive drug exposure.”

We interpreted the following Questions 3-9 from the reviewer as an element of the question raised in Question 1 about better validating that this approach is influencing cocaine-seeking through exposure to cocaine and not a constitutive effect of expressing the ion channel.

3. Although not statistically significant in the overall ANOVA, there was a clear trend for rates of food responding to be lower in the coca-5HT3 rats relative to control rats, even though they are cocaine-naïve. This is particularly apparent during the initial 1-5 days of training (under the FR1 schedule) but the trend persisted throughout testing. This could suggest that expression of coca-5HT3 in the habenula impacts operant responding independent of cocaine treatment. Thus, a broader range of behavioral assessments should be performed to confirm that coca-5HT3 expression in the habenula does not impact behavior under baseline conditions.

As noted by the reviewer, this was not statistically significant, but we understand why the issue is being raised because of the importance to verify the absence of a cocaine-independent effect of coca-5HT3. Thus, we prepared a new set of rats expressing coca-5HT3 in the LHb and performed sucrose consumption preference tests and operant sucrose self-administration tests. In both experiments there was no statistically significant effect of expressing coca-5HT3 (Extended Data Fig. 8). This data further supports the conclusion that cocaine-independent reward-guided behaviors were not affected by coca-5HT3, consistent with the cocaine-dependent mechanism of action of coca-5HT3 such that it only leads to negative feedback control of reward seeking during cocaine self-administration.

4. Related to the above, baseline electrophysiological properties of coca-5HT3-expressing habenula neurons should be assessed (resting membrane potential; spontaneous action potential frequency; rheobase, etc.).

We added data on the baseline electrophysiological properties of LHb neurons with coca-5HT3 and control LHb neurons in Extended Data Fig 7. No statistically significant differences were observed.

5. Likewise, the effects of coca-5HT3 on cell responses in the RMTg and/or other brain regions that receive synaptic input from the LHb should be assessed.

The connection of the LHb to the RMTg is established by many prior studies, and we cite this evidence in the paper (citations 40-45). Moreover, it is a difficult experiment to characterize chemogenetic modulation of this LHb synaptic input to the RMTg. This is largely because ion channels based on the $\alpha 7$ -5HT3 scaffold, such as coca-5HT3, are primarily expressed in the somatodendritic compartment and not axon terminals, and this precludes the use of brain slice electrophysiology due to the large physical separation of LHb and RMTg. Because our study is focused on the development of cocaine-gated ion channels and the proof-of-concept demonstration of negative-feedback chemogenetics, we felt that it was not crucial to show the already demonstrated LHb→RMTg connectivity with the only addition being to use coca-5HT3 for this (especially because coca-5HT3 is not well-suited for this due to its subcellular distribution).

6. Are there any changes in the self-administration of drugs of abuse other than cocaine (nicotine or heroin, for example) in coca-5HT3-expressing animals? Likewise, does coca-5HT3 expression alter responding for a high-value natural reinforcer like sucrose? This is important to confirm drug/behavioral selectivity.

We now address this question for sucrose in Extended Data Fig. 8. As before, there was no statistically significant effect of expressing coca-5HT3 in the LHb on natural reward seeking or consumption. Regarding IVSA for other drugs, we have already shown that coca-5HT3 is not responsive to opiates in Extended Data Fig. 4b and nicotine has potency for coca-5HT3 >100-fold right shifted to maximum self-administered concentrations in vivo (<0.5 mM). Redacted We hope the reviewer agrees with our assessment that the additional data regarding natural rewards coupled with in vitro data on the responses of coca-5HT3 to other addictive drugs should be sufficient to address the core concern about whether coca-5HT3 is specific to a cocaine reward.

7. Habenula inactivation is known to induce hyperactivity. Cocaine also induces hyperactivity. Are coca-5HT3 animals so hyperactive that they are unable to respond for cocaine?

We tested rats expressing coca-5HT3 in LHb neurons in an open field experiment in the absence of cocaine. There was no significant difference in locomotor activity relative to controls, showing that expression of coca-5HT3 did not introduce baseline hyperactivity (Extended Data Fig. 9). Related to this, for Question 9, we found that baseline dopamine was not significantly different between these rats, consistent with the notion that coca-5HT3 does not cause a basal perturbation. This is in contrast to the effects on cocaine self-administration. This and the previously requested control experiments further support our claim that the reduction of cocaine self-administration and cocaine-induced dopamine release is related to the cocaine-induced feedback mechanism that is inherent to the design of these ion channels. We also note that it has been previously shown that chemogenetic activation of the LHb decreases cocaine-induced hyperactivity (PMID: 31994279), thus we would not expect hyperactivity to be a cause of the reduced cocaine-seeking in the negative feedback chemogenetic experiment.

8. Other addiction-related behaviors known to be modulated to cocaine and that may involve habenula neurons should be assessed. For example, is the stimulant effect of cocaine altered, or cocaine-induced locomotor sensitization?

We appreciate the interest in this, however, we have not performed these experiments at this time because it involves characterization of new roles of LHb using the negative feedback chemogenetic approach. This would require additional investigation that goes beyond the focus of our manuscript on the engineering of these new ion channels and validation of the negative feedback chemogenetic concept. Moreover, we think that our focus on self-administration is the most important readout, especially relative to sensitization. This is because sensitization studies typically involve intermittent, low-dose, and non-contingent drug exposure, unlike addiction related behaviors. Furthermore, both humans (PMID: 35151770) and nonhuman primates (PMID: 34645982)

generally do not exhibit sensitization of physiological and behavioral drug effects; instead, tolerance is more commonly observed. Also, the mechanisms underlying locomotor sensitization often differ from those driving drug self-administration. For example, while glutamate receptor antagonists block the development of locomotor sensitization, they have limited effects on opioid or psychostimulant self-administration. Given these limitations, we chose to focus on the self-administration model in this project. Drug self-administration is the gold-standard method for evaluating a drug's addiction potential and testing potential addiction treatments.

9. The PET imaging and GRABDA photometry are interesting. However, but are “relative” measures of dopamine signaling. It is important to confirm that coca-5HT3 expression in the habenula does not impact “absolute” dopamine levels in the accumbens.

To address this critique, we measured the concentration of dopamine using ELISA in dissected Nucleus Accumbens tissue (Extended Data Fig. 11). There was no significant difference in basal dopamine levels between coca-5HT3 and control rats. We note here that we did not expose these rats to cocaine so that we would not introduce potential effects on basal dopamine associated with cocaine-exposure.

10. What are the effects of coca-GlyR on cocaine-related behaviors and dopamine signaling in the accumbens?

This question is similar to a request by Reviewer 2 (their comment #9) and is related to providing some in vivo validation of the effect of this inhibitory channel. We had access to a breeding colony of Th-Cre rats at NIDA, so we focused on the more direct measurement of cocaine-induced increases in extracellular dopamine using [18F]-fallypride and PET imaging. In these new experiments, we found that rats expressing coca-GlyR in TH-neurons in the VTA showed reduced displacement of [18F]-fallypride relative to control rats receiving cocaine, which demonstrated reduced extracellular striatal dopamine in the presence of cocaine for the VTA-Th-coca-GlyR rats.

Referee #2 (Remarks to the Author):

This is an elegant study that describes the approaches used to generate and utilize a new tool that will be highly beneficial to investigators in the drug dependence field. The authors provide an excellent series of experiments that shows the characterization of these chimeric cocaine-selective receptors and then demonstrate their utility in in vivo models. Altogether, this is exciting work. That said, there are some issues that need to be addressed. Much of these issues are likely due to writing within the word limit of the journal. Therefore many of these issues are associated with clarity of the specific methods.

Major Comments:

1. Page 4: The authors reference the pharmacophore of shared similarity between nAChR agonists and addictive drugs but Extended data 1b looks to have not made it into the submission (only the chimeric schematic of extended data fig. 1a is present, and it does not include 1b).

Thank you for pointing this out. We have corrected it and Extended Data Fig. 1b is now included.

2. Page 6, last paragraph: The authors used viral methods to express Coca-5HT3 for electrophysiology

assays. However, they do not discuss if these are cultured neurons or in slices (the methods do specify hippocampal cultures but this should be clear in reading the Results section).

We have added information about the methods for electrophysiology in cultured hippocampal neurons and (new data) LHb slices. We also added this in the Results.

3. Why were rheobase measurements not completed for coca-5HT3 as they were with coca-GlyR?

We have added this to Fig. 2e. Rheobase is reduced for coca-5HT3 in the presence of cocaine, as expected for a depolarizing channel.

4. Figure 2d shows a concentration-dependent increase in neuronal firing. Why is there no mean data associated with this observation? Were these current clamp recordings not replicated in a manner that would facilitate an analysis of the mean?

These current clamp recordings are shown for illustrative purposes but there is generally variability in the firing properties of neurons to pharmacological depolarization based on resting membrane potential, membrane resistance, cell size etc. The consistent response of depolarization in these cultured neurons is what we emphasize in Fig. 2c and this corresponds well to the cocaine EC50 of the channel. Fig. 2d is an example of firing that is related to this type of depolarization but is expected to be variable due to above factors that are not directly related to the effectiveness of the channel to depolarize neurons. Now that we added data for the rheobase in Fig. 2e, the group effect on excitability due to this depolarization is better illustrated.

In 2c, the authors show mean data for cocaine-induced depolarization of hippocampal neurons but don't provide representative traces. This also occurs with Figure 2e. In both cases, representative waveforms should also accompany the mean data. If there is not sufficient space, then this could be put in the extended data.

In this part of the question, the reviewer is pointing out the lack of representative traces for coca-GlyR. In the new figure 2, we have added these traces in Fig. 2e,f and the dose responses from the former Fig 2e are now moved to Fig. 2h. We have added voltage clamp recordings for the Cocaine and ACh responses of coca-GlyR in Fig 2f and 2g. Note that the difference in activation timecourse for coca-5HT3 and coca-GlyR is consistent with all our (and others) past work on these chimeric channels, with a7-GlyR chimeric channels always showing slow activation kinetics, which was investigated previously (PMID:16319224).

5. The only photometry data present is the processed Z-scores of control and coca-5HT3-injected rats. The authors should provide sample data of raw signal and isosbestic signal traces. Many investigators inject GRABDA or dLight and then implant optical cannulas. Given that the authors implanted cannulas that were coated in GRABDA virus, it would be very important to show what the raw photometry signals look like. Given that the authors used workflows from the Lerner lab (GuPPy perhaps?), some of these data will have plots generated already and these can be added to a revised figure.

Representative traces are added in Extended Data Fig. 14.

6. The results section associated with the photometry is also vague. How was the cocaine treatment

conducted? Was this done in a simple open field, rats were allowed to habituate, and then the rats were sc/ip injected with cocaine and placed back in the arena during the recording session? Once again, much of this is explained in the methods but some of these details should be provided in the results section. This also highlights the need for providing traces as there were likely handling artifacts for injecting the cocaine during the photometry sessions.

We have added additional details in the results on p. 11: “Rats were anesthetized and their optical fiber was connected to the patch cord. GRAB_{DA} signals were then collected for 5 minutes, followed by a saline i.p. injection, and additional GRAB_{DA} signal measurement for 20 minutes. Cocaine was then injected i.p. at either 20, 10, or 5 mg/kg, and GRAB_{DA} signal was measured for an additional 20 minutes. Each rat was tested 3 times (starting at 20, then 10, then 5 mg/kg) in randomized order.”

7. Specific for the photometry measurements, the authors should provide the frequency at which the recordings were set. The laser power was provided but it was not clear if this was for both the GRABDA signal and for the isosbestic.

The information has been added to the methods: “Data were acquired at 1 kHz.”

8. No electrophysiological methods were provided for the recordings of the hippocampal neurons. I assume the same amplifier/digitizer setup was used as with the HEK293 electrophysiology. However, with cultured neurons some investigators use specific extracellular buffers that differ from what is used for HEK cells. This should be provided. Also, there needs to be a discussion of what protocol was used to measure Rheobase. Finally, the authors should describe what methods they used to calculate specific cell parameters (capacitance, resistance, input resistance, and resting potential).

The information has been added to the methods.

9. The in vivo utilization of coca-5HT3 provides functional validation for these chimeric proteins in a behavioral model. However, the authors developed and characterized the coca-GlyR protein as well. As the functional validation shows, cocaine will activate the coca-5HT3 receptors robustly but will inactivate the coca-GlyR. Given the behavioral result of ‘selective’ activation of coca-5HT3 in the LHb, it would be extremely interesting to see if using coca-GlyR to impose the opposite effect would produce the reverse. Understandably, this would take significant time to recapitulate the cocaine IVSA with coca-GlyR but something that may be less time intensive would be to see if coca-GlyR would enhance NAc dopamine given that coca-5HT3 depressed NAc dopamine release.

Based on this question, we designed an experiment to test the effect of coca-GlyR expressed in Th-Cre neurons of the VTA on cocaine-induced dopamine release in the ventral striatum. Cocaine-induced suppression of VTA dopamine neurons by coca-GlyR was expected to reduce dopamine release and result in lower extracellular dopamine in the striatum which was measured with PET imaging using displacement of [18F]-fallypride. Here, we found that rats expressing coca-GlyR in TH-neurons in the VTA and injected with cocaine showed reduced displacement of [18F]-fallypride relative to control rats receiving cocaine. Lower [18F]-fallypride displacement in coca-GlyR rats demonstrates suppression of dopamine release in response to cocaine as compared to control rats.

Minor Comments:

1. Figure 3C: The number of days (or range in days) for food training, cocaine training, and dose response should be indicated in the schematic

We have modified this figure as suggested.

2. Figure 4D: The authors should add a legend to describe the different data points.

We have modified this figure as suggested.

3. Page 4: “ α 7 nicotinic acetylcholine receptor (nAChR) ligand gated...” Is not the typically language used within the nAChR field. I know this was just a stepping stone to generate the protein(s) at the focus of this work. However, it would be more correct to rework this sentence to something like: “ α 7 nicotinic acetylcholine receptors (nAChRs) are ligand gated ion channels that have been engineered...”

We have modified this sentence in a manner similar to what was suggested.

4. Page 5: references for ACh/nAChR cation- π binding should include additional references since this has been replicated in multiple labs and model systems. The original work that led to our current knowledge dates back to 1998 (Zhong et al., 1998, PNAS).

Excellent point. We apologize for this mistake in not citing the primary studies. We have corrected this.

5. The authors show competition binding of cocaine for the coca-5HT3 receptor. Was the same assay completed for any of the α 7GlyR chimeric mutants?

We have now performed this experiment, and we found that coca-GlyR shows a binding curve that is best explained by a two-site binding model, suggesting a high and low affinity state of the receptor in these experiments. We have included this data in Extended Data Fig. 6.

This result led us to go back and re-test coca-5HT3 with a higher resolution dose response curve for cocaine and ASEM. We found a similar effect with coca-GlyR. Because of this, we also replaced the binding curves in Fig 1g for ASEM and cocaine at coca-5HT3 that were previously based on a one-site model with the new two-site model that fits more cocaine and ASEM concentrations.

6. In the legend of Figure 2(d) the author state that current injections were used to monitor membrane properties during the collection of current-clamp data. This needs to be detailed in the methods or extended data.

We added this to the methods on p.21: “Membrane properties during current-clamp recordings were monitored by small current injections (-20 pA, 200 ms, 1 Hz).”

Referee #3 (Remarks to the Author):

Gomez et al developed novel engineered ionotropic receptors that respond selectively to cocaine, and use

them to demonstrate what they refer to as “closed-loop” control of cocaine self-administration via manipulation of lateral habenula circuits influencing nucleus accumbens dopamine signaling. The paper offers two main contributions to the field.

The first is their strategy for engineering a drug (cocaine)-sensitive excitatory or inhibitory ionotropic receptor, and validating its function in vitro and in vivo to control certain self-administration behaviors, and downstream dopamine signaling.

The second and more important is the concept of ‘closed-loop’ control of neural circuits underlying cocaine intake. The idea of closed-loop control of maladaptive neural circuit activity is not novel per se, but to my knowledge this is the first test of the idea in the context of intake of a specific drug, here cocaine, and thus it represents a conceptual advance. This new idea could be experimentally employed to test circuit functions and control of behaviors in animals as done here, and it could potentially in future be implemented to control maladaptive behaviors in humans addicted to specific drugs like cocaine. They note this closed-loop approach could also be developed to intervene in other biological feedback systems such as those involving hormones or nutrients, which is an exciting potential future application.

However, the paper has some significant limitations as well.

The premise of closed-loop control of cocaine intake relies here upon a neural pathway (LHb midbrain GABA populations dopamine neurons) that is assumed, but not actually directly tested. It was surprising that given the focus on dopamine neurotransmission as an endpoint, the cocaine-sensitive receptor was not expressed in dopamine neurons themselves—this would be more direct test of the proposed mechanism of “closed loop control”.

We also answer this in Question 5 for Reviewer 1. The circuit of LHb→RMTG+VTA dopamine neurons has been established by past work. We cite these papers in the text (citations: 40-45). We want to emphasize that the significance of our work is to engineer the first cocaine-gated ion channels and to apply them as closed-loop chemogenetic tools. We show that they work by suppressing cocaine-induced striatal dopamine release to reduce cocaine seeking. In new experiments, we show that by expressing coca-GlyR in VTA dopamine neurons, we can reduce the cocaine-induced rise in striatal dopamine (ED Fig. 12).

Other issues exist regarding specificity of the presented data. They show that the receptor does not bind other addictive drugs or endogenous neurotransmitters, but drugs structurally similar to cocaine, some of which are used clinically like lidocaine or Novocain, were not tested.

To address this critique, we tested Procaine (Novocain) and Lidocaine, which did not activate coca-5HT3 up to 100 μM. Because the reviewer asked about clinically used drugs that are structurally similar to cocaine, we also tested drugs that contain a tropane pharmacophore (like cocaine), which would be most likely to activate the ion channel. These data are in Extended Data Fig. 5b. The only drugs with a substantial response between 10-100 μM were Benztropine and Methyaltropine. Interestingly, 8 other tropane containing drugs did not show substantial activity. Although we don't think this activity has substantial implications for the application of these chemogenetic receptors, we note that methyaltropine is primarily used for ocular applications and has poor brain penetrance. Benztropine (used as an anti-psychotic or movement disorder treatment) would be likely contraindicated for any research or potential clinical application of coca-5HT3 (although this would still depend on actual brain exposure levels in clinical treatment regimens).

Anatomical specificity of presented behavioral and dopamine data should also be clarified. Does, as one would expect under the proposed framework, intra-LHb cocaine do the same things as systemic cocaine?

Although we would expect intra-LHb cocaine to activate the coca-5HT3, we need to give systemic cocaine to test the specific role of reversing the cocaine-induced LHb neuron inhibition on cocaine-seeking behavior. Thus, it is unclear that we could evaluate how it would modulate behaviors relevant to cocaine reinforcement and addiction.

Does cocaine (systemic or intracranially applied to the receptors) alter other behaviors controlled by LHb and its projections?

Because the purpose of this study is to engineer the first cocaine-gated ion channels, to apply them as closed-loop chemogenetic tools and to characterize their effect on drug-seeking, we felt that exploring other roles of the LHb was outside this scope. As we replied to Reviewer 1, we think these are the type of experiments that should be pursued in new research after the description and characterization of these new channels is reported as well as the initial validation of negative feedback chemogenetics. Redacted

The main behavior associated with LHb activation that is relevant to cocaine-seeking is the established role of the LHb to reduce effortful responding (PMID: 29752382). This is consistent with the reduced drug-seeking we observe when high levels of effort are required to maintain a reinforcing dose of cocaine. In addition to this, we did perform additional new experiments to further show that coca-5HT3 in the absence of cocaine was not leading to basal perturbation of dopamine or cocaine-independent behaviors (Extended Data Figs. 8, 9, 11).

Does PET show cocaine at relevant doses demonstrably binds the receptor in vivo?

Our data indicate that cocaine and ASEM bind to coca-5HT3 and coca-GlyR in a biphasic manner, which would introduce nonlinearity into tracer-binding kinetics and complicate quantitative interpretation of PET data as the tracer binding may not be proportional to receptor availability. Therefore, it is not likely that [11C]cocaine or [18F]ASEM could be used to label coca-5HT3 or coca-GlyR reliably in vivo, especially given the limited regional expression in the LHb, and binding of [11C]cocaine and [18F]ASEM to endogenous DAT, NET, SERT, and $\alpha 7$ nAChR respectively.

Another feature of the paper that I have trouble with is the pattern of effects on cocaine self-administration behavior. Control and coca-5HT3 mice acquire self-administration in a “statistically equivalent manner,” which is hard to understand given the proposed “closed loop” mechanism—wouldn’t cocaine in these animals be expected to attain an aversive quality that would suppress self-administration? Lines 177-80, and 261-63 address this point and note that “coca-5HT3 does not significantly interfere with cocaine self-administration with a high infusion dose that does not require high instrumental responding to maintain exposure to reinforcing levels of cocaine”. However I do not understand this logic since effort required to obtain cocaine should not be especially relevant to the rewarding vs aversive properties of cocaine itself—the variables which are proposed to interface with cocaine intake in a closed-loop manner. Likewise, since only high effort cocaine intake is affected by the manipulation, and high dose cocaine intake is not affected, this seems to undercut the translational relevance of the approach, since an addicted human could override the intervention by taking a larger dose of cocaine.

Acquisition of cocaine self-administration requires high unit doses to produce sufficient pharmacodynamic effects for rats to form a strong conditioned association between cocaine, the lever, and the cocaine-paired cues. If the acquisition unit dose is too low rats do not acquire self-administration (PMID: 9084058). The acquisition unit dose of cocaine in our study was 0.5 mg/kg/infusion. This unit dose is on the descending limb of the cocaine unit dose-response curve. Whereas such high unit doses facilitate acquisition of self-administration, they are not useful for evaluating the effects of pharmacotherapies on drug self-administration. This is because they limit rates of responding via non-reinforcing behaviors or mechanisms (e.g., stereotypy, hyperlocomotion) (PMID: 8726752).

From an addiction treatment perspective, the unit dose-response curve is used for evaluating the efficacy of a treatment in decreasing cocaine self-administration (PMID: 8726752). For a treatment to be successful, it should aim to reduce cocaine responding along the unit dose-response curve specifically at doses where high levels of self-administration are maintained. A depression in the ascending limb of the dose response curve, which is what we observed, would mean that the treatment decreases the efficacy of cocaine, reducing the drug's maximum response and overall magnitude of reinforcement or reward without affecting non-specific mechanisms. In contrast, a right or left-shift in the dose-response curve as seen in another publication (PMID: 9860114), which we did not observe, would indicate that the pharmacological treatment antagonizes or potentiates cocaine reinforcement respectively, resulting in a decrease or increase in cocaine potency. It is the rightward shift in the dose-response curve because of a decrease in cocaine potency, not efficacy, that could allow one to override the pharmacotherapy by self-administering a larger unit dose.

We give the following explanation in the Discussion on p. 13: “LHb activation did not interfere with the motor actions of cocaine seeking or the capability of cocaine to reinforce these actions at high doses, because rats learned to self-administer reinforcing doses of cocaine. Instead, artificial cocaine-mediated LHb activation selectively suppressed the higher response rates necessary to maintain reinforcing exposure to cocaine at lower dose infusions. This was likely due to a reduction in the release of dopamine in the nucleus accumbens.”

More explicit information about the sex of experimental subjects in each experiment is also required—some appear to be only in males despite the information provided in the additional information provided for review saying all studies are in males and females.

We apologize for this discrepancy. Only male rats were used. We corrected this in the supplemental information.

Finally, the initial sections of the paper would benefit from concrete definition of conceptual terms such as “non-physiological open-loop neural perturbations”.

We removed the phrase “non-physiological”. Open-loop is defined in the subsequent clause as not being coupled to addictive drug concentrations in the brain. This is mentioned as a contrast to our development of a closed-loop chemogenetic approach by making the chemogenetic perturbation intrinsically dependent on cocaine levels in the part of the brain with the receptor.

Response to Reviewers

Reviewer 2:

- 1. We made the error bars consistent across the panels**
- 2. We stated the statistical test in the Figure 3 legend**
- 3. We stated the statistical test in the Figure 4 legend**

Reviewer 3:

Regarding concerns about the claims of a negative feedback mechanism and at the editor's suggestion, we have altered the title and the phrasing to remove negative feedback from the manuscript.

Minor Points:

Line 261: "Each rat was tested 3 times (starting at 20, then 10, then 5 mg/kg) in randomized order". Clarify if doses were tested in fixed or randomized order.

We state they are tested in randomized order

-Line 552-3: "The tissue was then chunked removing the olfactory bulbs and coronally sectioning at the optic chiasm." For ELISA dopamine analysis: clarify what this means, and which tissue exactly was analyzed for DA levels

We added this information in the methods: "The tissue that contained the bilateral striatum was then homogenized and analyzed for dopamine levels using ELISA (abcam, ab285238)."

-Line 590: "Rats could commit a maximum of 60 infusions when tested at the 0.5 and 1 mg/kg unit doses but no infusion limit was present at other unit doses" Authors should note if any rats reached the 60 infusion limit, and any potential implications of this on the data overall.

We note that the rats did not reach this limit

-Please clarify for all experiments what strain and sex of animal was used, and acknowledge in the manuscript any potential implications of the multiple rat strains used, and exclusive use of males, on implications and generalizability of these findings.

We note the limitation of using only males and using two rat strains in the Discussion: "One limitation of this study is that it focused solely on male rats and included

use of two different rat strains. Further investigation is needed to examine the extent to which these effects generalize to female rats and other rat strains or species.”

-Please clarify in reporting checklist that females were not used in any experiments (or where they were used if they were).

This is noted

-Clarify in main text that vectors targeting dopamine neurons in TH:Cre rats were cre-dependent
We have added this information to the main text